

# Controls on spatial and temporal patterns of soil nitrogen availability in a High Arctic wetland

Jacqueline K.Y. Hung[1], David M. Atkinson[2], Neal A. Scott[1]

[1]Department of Geography and Planning, Queen's University, Kingston, K7N 3N6, Canada

[2]Department of Geography and Environmental Studies, Ryerson University, Toronto, M5B 2K3, Canada

*Correspondence to*: Jacqueline K.Y. Hung (**jacqueline.hung@queensu.ca**)

**Abstract.** Increased soil nutrient availability, and associated increase in vegetation productivity, could create a negative feedback between Arctic ecosystems and the climate system, reducing the contribution of Arctic ecosystems to future climate change. To predict whether this feedback will develop, it is important to understand the environmental controls over

nutrient cycling in High Arctic ecosystems, and how they vary over space and time. This study explores the environmental controls over spatial patterns of soil nitrogen availability in a High Arctic wet sedge meadow and how they influence carbon exchange processes. Ion exchange resin membranes measured available inorganic nitrogen in soils throughout the growing season at a high spatial resolution, while environmental variables (e.g. active layer depth, soil temperature, soil moisture) and carbon flux measurements were taken at frequent intervals during the 2016 field season. Environmental measures correlated

highly with total and late season nitrate levels (total season dry tracks nitrate $R^2 = 0.533$, total season wet tracks nitrate $R^2 = 0.803$, late season nitrate $R^2 = 0.622$), with soil temperatures at 5 cm depth having the greatest effect. Soil available nitrate and ammonium correlated highly with total and early season gross primary productivity (total season wet tracks $R^2 = 0.685$, early season dry tracks $R^2 = 0.788$, early season wet tracks $R^2 = 0.785$). Higher ammonium concentrations coincided with greater carbon dioxide uptake. Nitrate concentrations correlated strongly to soil moisture, but nitrate levels were much lower

than ammonium concentrations, suggesting low rates of nitrification vs. mineralization. Similar patterns were observed regardless of whether the wet-sedge meadow was classified as wet or dry, but the relationships were always stronger in areas classified as wet, indicating the importance of moisture and water availability on abiotic processes in High Arctic wet sedge meadows. Topography played an important role in the movement and transport of water, which influenced how nutrients were cycled and moved within the wetland. Generally, the low-lying areas had the highest inorganic nitrogen concentrations.

These results suggest that finer scale processes altering nitrogen availability may influence the overall carbon balance of wet sedge meadows in the High Arctic, and how these ecosystems may respond to changes in climate.

# 1 Introduction

Warming temperatures in the High Arctic are expected to exceed global rates by 40% (IPCC, 2013). Changes in seasonal weather patterns may also influence ecosystem-level abiotic factors, which in turn will influence biogeochemical

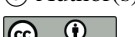



processes (e.g. carbon (C) and nitrogen (N) cycles) in complex ways. Some of the consequences of Arctic climate change include increased air temperatures leading to earlier snowmelt onset (Young, 2006) and precipitation increase leading to permafrost degradation and the release of previously unavailable soil C (Oechel *et al.*, 1993; Schuur *et al.*, 2009). Warming is expected to accelerate the decomposition of soil organic matter (SOM) (Chapin *et al.*, 1995; Aerts *et al.*, 2005; Bell *et al.*, 2013), potentially altering rates of nutrient cycling. N fixation by key ecosystem species like *Nostoc commune* and *Sphagnum spp.* will also change with air temperature increases (Chapin *et al.*, 1995; Schmidt and Lipson, 2004), potentially altering available nutrient pools. In turn, higher nutrient availability could alter species composition within High Arctic ecosystems (Aerts *et al.*, 2005), plant productivity (Shaver *et al.*, 2000), and the net C balance (Welker *et al.*, 2004). Some of the changes to Arctic regions are already being seen with amplified warming (Bintanja and Andry, 2017) and further changes are expected. These changes could lead to the development of negative feedbacks, potentially helping stabilize the climate system. Given the number of changes and potential responses, an ecosystem-level understanding of these interactions is critical to understand how they will respond to future changes in climate.

High Arctic plant growth is typically limited by N availability (Nadelhoffer *et al.*, 1992; Shaver and Chapin, 1995; Shaver *et al.*, 2000). As such, this study looked at the spatial patterns of plant-available inorganic N to see how these patterns shift during the growing season. A wetland was investigated in this study for its characteristics as the most productive and photosynthetically active ecosystems in the High Arctic environment (Atkinson and Treitz, 2013). High Arctic wetlands play an important role in the hydrology and C dynamics of Arctic environments (Grogan and Jonasson, 2005). Wetlands require a constant, sufficient water supply and are found where water gains exceed losses (Woo, 2011). Consequently, the availability of water in the otherwise semi-arid environment of the Arctic is a determining factor in the location and productivity of these wetlands (Woo and Young, 2006; Woo and Young, 2012). In the Arctic, their growing season is limited to a maximum of three months each year, during which an adequate supply of water is needed to sustain them as wetlands. These moist areas provide sustenance for Arctic fauna and play an important role in ecological function (Woo and Young, 2006; Woo and Young, 2012). With changes in climate anticipated in the High Arctic, land classes such as wet sedge meadows will likely respond dramatically, and changes in these wetlands will have cascading effects on terrestrial and hydrological features around them.

Arctic wetlands act as important regions of long-term C storage and sequestration. These wetlands contain a disproportionate amount of subterraneous C (Post *et al.*, 1982; Grogan and Chapin, 1999) and could contribute significantly to global C changes. It is expected that increased C losses from the terrestrial environment into the atmosphere will come as a result from permafrost degradation due to the changing global climate (Tarnocai *et al.*, 2009). Preliminary research has predicted that Arctic wetlands in Arctic regions have the potential to increase C outputs with increased temperatures and precipitation inputs (Nobrega and Grogan, 2008; Hill and Henry, 2011). These increases in wetland biomass can help offset the projected increases through C uptake during photosynthesis. However, global climate models by Avis *et al.* (2011) have also projected the opposite, with permafrost degradation leading to a decrease in the areal extent of wetlands. As such, more





research is needed to understand the relationship between climatic warming and Arctic wetlands to determine how they will respond to increased temperatures.

There has been a 40% increase in atmospheric $CO_2$ concentrations since pre-industrial times (1920-2017), and concentrations continue to increase each year (IPCC, 2013). Comprehensive data from NOAA's Barrow station indicated
that early winter $CO_2$ emissions have risen 75% since 1975 (Commane *et al.*, 2017). Arctic tundra soils contains 14% of the global C pool (Post *et al.*, 1982; Grogan and Chapin, 1999) and their potential release into the atmosphere could influence global climate. Nobrega and Grogan (2008) have found in their study of mid-Arctic wetlands that net C gain was largest in an Arctic wetland setting when compared to other Arctic ecosystems. Short-term studies of Arctic tundra environments are suggesting that Arctic permafrost regions currently act as sinks of atmospheric and terrestrial C (Nobrega and Grogan, 2008;
Lafleur *et al.*, 2012; McGuire *et al.*, 2012); however, the comprehensive C study by McGuire *et al.* (2012) also determined that the tundra has been neutral in recent decades. Generally, long-term C studies are still lacking across the High Arctic tundra (Euskirchen *et al.*, 2016). A long-term study by Euskirchen *et al*. (2016) found that increases in air and soil temperatures at multiple depths may trigger a new trajectory of $CO_2$ release. As such, the need to understand the environmental variables that influence these processes is necessary, particularly in ecosystems like wetlands that have a
disproportionate influence on the landscape C balance in the High Arctic.

The balance between ecosystem respiration (ER) and gross primary productivity (GPP) is an important determinant of net C storage. High Arctic wetlands have long been regarded as C sinks due to the dominance of GPP over ER (Mikan, *et al.*, 2002; Nobrega and Grogan, 2008; Lafleur *et al.*, 2012; McGuire *et al.*, 2012. However, experimental warming in the Alaska Range has shown that organic C near the surface of these wetlands is more vulnerable and susceptible to release with
warming, creating a positive feedback (Post *et al.*, 1982; Natali *et al.*, 2011; Commane *et al.*, 2017). Warming temperatures are expected to increase ER in the Arctic, which could lead to a shift in the C balance depending on the response of GPP; this could lead to decreased net C storage and make permafrost regions long-term net C sources (Welker *et al.*, 2004; Commane *et al.*, 2017). Belowground $CO_2$ release is season-dependent and strongly influenced by climate (Grogan and Chapin, 1999). Historically, the Arctic's frozen soils were considered strong C sinks due to low temperatures and poor
drainage (Grogan and Chapin, 1999; Mikan, *et al.*, 2002). However, some Arctic areas like the Alaskan North Slope have consistently been acting as C sources during the growing season (Oechel *et al.*, 1993). More recently, studies have determined that the Alaskan Arctic has been a net $CO_2$ source from 2012 to 2014, with particularly high emissions in the early winter months (Commane *et al*., 2017). Similarly, a study that included a Canadian High Arctic wetland at Alexandra Fiord on Ellesmere Island showed that the wet sedge was a net $CO_2$ source until switching to a sink closer to the end of the
growing season (Welker *et al.*, 2004). Studies at Toolik Lake, Alaska found that simulated warmer climate made wet sedge tundra plots a weak sink for $CO_2$ at the peak of the growing season but only for a short period of time (Johnson *et al.*, 2000). The study of C cycling in Arctic wetlands by Sullivan *et al*. (2008) found C fluxes to be temperature-sensitive, concluding that with future warming trajectories. High Arctic wetlands may respond more rapidly due to their smaller temperature ranges. Most of the previously mentioned studies did not consider how changes in nutrient availability might influence



ecosystem C balance. Bunnell *et al.* (1977) quantified the temperature response of microbial respiration ($Q_{10}$) in the Arctic as 3.7 for soils between 0 and 10°C (Robinson, 2002), which is the average temperature range of wet sedge soils in the growing season. When Billings *et al.* (1984) increased soil N through fertilization, a significant increase in $CO_2$ uptake was recorded in the harvested tundra soil cores. Past research has shown that a relationship exists between available soil N and measured

environmental variables; this study aims to determine the spatial and temporal extents of these relationships in the CBAWO wetlands.

The nutrient regime in Arctic plant growth is typically characterized by slow growth rates that are nitrogen and phosphorus-dependent (Jonasson and Shaver, 1999). However, relative to more active, nutrient-rich temperate environments, tundra environments are more responsive to short-term (1-10 year) changes in nutrient availability (Shaver *et al.*, 2000). Due

to the interconnected nature of the physical and microbial processes of Arctic wetlands, it is expected that changes in nutrient availability will be reflected in processes such as C flux and denitrification. In tundra microcosms (Billings *et al.*, 1984), increased soil N significantly increased $CO_2$ uptake. Future climate changes are expected to increase soil temperatures (ST) which will deepen active layers, lowering the water table, promoting subsurface flow pathways and water availability that allows for nutrient release from decomposing organic matter (Biederbeck and Campbell, 1973; Billings *et al.*, 1982;

Nadelhoffer *et al.*, 1991; Johnson *et al.*, 2000; Natali *et al.*, 2011). These factors in turn affect other abiotic characteristics such as soil moisture (SM) and pH. Changes to these environmental variables will affect the processes and dynamics within Arctic soils, including key processes like N cycling. Plant-available N in the forms of $NH_4^+$ and $NO_3^-$ is a major limiting factor for growth in the High Arctic (Nadelhoffer *et al.*, 1992; Shaver and Chapin, 1995) and concentrations are usually relatively low. However, the factors causing this shortage of plant-available N are not fully understood (Beermann, 2016)

and the circumpolar N pool needs to be further investigated. Larose *et al.* (2013) found N cycling to be sustainable in water-limited environments like the base of snowpacks through the work of microorganisms. The microbial communities were able to shift their functional potential to allow for several pathways of the N cycle to continue despite the low temperatures and limited water (Larose *et al.*, 2013). These processes may be important in the wet sedge meadow of the study presented here, which is located at the base of a perennial snowpack. Research from field studies has also shown soil decomposition by

microbes may be limited by N (Mack *et al.*, 2004; Lavoie *et al.*, 2011). Understanding the role N plays in Arctic ecosystems will help in future predictions of decreasing soil C storage and microbial decomposition as the environments continue to change (Chapin *et al.*, 2002; Mack *et al.*, 2004)

Microbial controls on nutrient cycling are an important processes to consider in High Arctic environments. Mikan *et al.* (2002) found that warming in laboratory incubation studies stimulated microbial activity and increased nutrient turnover

in thawed soils. Microbial activity is known to remain active throughout the winter season and can have significant contributions to nutrient budgets during spring thaw (Hobbie and Chapin, 1996; Schmidt and Lipson, 2004; Edwards *et al.*, 2006). Because the insulating snow layer prevents Arctic mid-winter soils from falling below -10°C (Clein and Schimel, 1995), the occurrence of freeze-thaw events allows microorganisms to remain active as long as pockets of liquid water are still present as a result of the snow insulation (Edwards *et al.*, 2006). The activity of these microorganisms will mobilize N



stores that can help mitigate the current nutrient limitation in High Arctic ecosystems (Mikan *et al.*, 2002), promoting plant metabolism and subsequent respiration and photosynthetic activity in the following growing season. In High Arctic wetlands where microbial metabolism is primarily anaerobic due to the anoxic conditions, changes to drainage, precipitation, or evapotranspiration patterns will be the primary driver of microbial activity changes in the future (Mikan *et al.*, 2002).

This study explores the spatial and temporal patterns of available soil N in a wet sedge meadow in the Cape Bounty Arctic Watershed Observatory of the Canadian High Arctic and how it relates to C exchange processes. Many factors determine the levels of plant-available N that fuel N cycling and C flux (Nadelhoffer *et al.*, 1991; Clein and Schimel, 1995; Chapin *et al.*, 2002), and the need to understand the relationship between the environmental variables and these processes is more important than ever in this climate change narrative. Previous studies in this wet sedge meadow have looked at the

seasonal variability of C flux and environmental variables and C fluxes across wet and dry areas within wet-sedge meadows. However, the relationships between available N and C exchange have not been assessed, and with a spatial lens. While nutrients have been examined in CBAWO wetlands, spatial and temporal extent of the sampling was much more limited than used in this study.

## 2 Methods

**2.1 Study Site**

The CBAWO (Fig. 1a) is located on the southern coast of Melville Island, Nunavut (74°54' N, 109°35' W), near the Nunavut-Northwest Territories border. The area contains two watersheds (East and West Lake), and covers approximately 150 km$^2$. It is in continuous permafrost with a 0.5 to 1-metre thick active layer (Atkinson and Treitz, 2013). The climate in the area is cold throughout the year with the summer melt and growing season running from June to August (Atkinson and

Treitz, 2013). January 2016 minimum and maximum temperatures were -33.7°C and -21.5°C respectively, while July 2016 minimum and maximum temperatures were 1.7°C and 10.9°C respectively.

The CBAWO was selected as the area of interest due to its availability as a High Arctic research area and the longevity of hydrological and biogeochemical research that has been conducted. This study was carried out from June 25 to July 28, 2016. The watershed is dominated by three land cover classes: polar semi-desert, mesic tundra, and wet sedge

meadows (Gregory, 2011; Atkinson and Treitz, 2012; Atkinson and Treitz, 2013). Wet sedge meadows in the High Arctic landscape tend to be in low-lying areas near a continuous water supply during the growing season (Woo and Young, 2006; Thompson and Woo, 2009; Woo, 2011). The wet sedge meadow of interest within the CBAWO, "Muskox", is a 200 metre by 200 metre plot that is saturated year round (Fig. 1b). Previous work conducted in this wetland examined carbon exchange (Atkinson, 2012; Gregory, 2011; Blaser, 2016). Water is provided to Muskox by a perennial snowpack located at the north

end of the area. Walker *et al.* (2005) give the Cape Bounty area a G2 vegetation classification of moist tundra covered with low-growing forbs, grasses, mosses, and lichens. The Muskox site slopes downward away from the snowpack from north to south as well as from east to west and has a south-facing aspect. Within Muskox, there are variations in dryness and wetness





across the sites: wet tracks are characterized by standing water in some areas and higher soil moisture, while dry tracks are still wet but lack pools of standing water. The partitions for these moisture tracks are small longitudinal hummocks that were likely formed from cryogenic events (Hodgson *et al.*, 1984).

The sampling area is dominated by *Carex*, *Gramineae*, and *Sphagnum spp.*, with the occasional bedrock outcrop.
The dwarf shrub *Salix arctica* and flowering *Eriophorum*, *Ranunculus rivalis*, and *Papaver radicatum* are also present in the wet sedge meadow. *Salix arctica* was most dominant in the wet tracks of the meadow, while herbaceous flowering plants tended to be found in dry tracks. Sharing the ground with the *Sphagnum* mosses were various lichen, including *Cladonia* and *Stereocaulon* genus. As the melt season progressed, the drainage of standing pools of water in the western portion of the meadow revealed the N-fixing cyanobacteria *Nostoc commune*.

**2.1.1 Experimental Design**

The sampling methodology was designed to sample variability in soil moisture within the meadow. The actual sampling sites were determined using *in situ* site reconnaissance with the aid of satellite imagery to discern between moisture tracks. A total of 64 sites were established on alternative wet and dry strips sampled using ion exchange resin strips; these sites represent an 8 row by 8 column grid that encompasses 4 wet and 4 dry transects within the meadow. The aim with this
sampling design is to test the effects of proximity to snowpack and its water input (e.g. differences in soil moisture) on nutrient availability within the wet sedge meadow, while giving insight into the spatial variability within the meadow.

**2.2 Soil Nitrogen Availability Evaluation**

Ion exchange resin membranes (Qian and Schoenau, 2005); were employed to determine soil N availability within the Muskox meadow at CBAWO. These resin membranes work by adsorbing nutrients that are attracted to the resident ions
on the resin membranes, which are placed onto the resins through pre-burial preparation techniques. Resin membranes were prepared using the ion exchange resin protocols developed at the University of Saskatchewan (Qian and Schoenau, 2005). Cation and anion exchange resins measuring 2.5 cm by 10 cm were made from membrane sheets obtained from Membranes International Incorporated (Membranes International Inc., Ringwood, New Jersey, United States). Cation and anion resins were pre-loaded with a 0.5M hydrochloric acid (HCl) solution for 24 hours with agitation to allow for the acid to strip the
resin of any ions currently present, after which they were soaked in 0.5M $NaHCO_3$ for 5 hours with agitation to saturate sites for sodium and bicarbonate ions. The anion probes were also soaked in a 0.01M ethylenediaminetetraacetic acid solution for one hour with agitation to help increase its adsorption of phosphorus. The resins were deployed at 10 cm soil depth remained in the field for the allotted burial time of two or four weeks. When removed from site, the resins from each experimental plot were cleaned and rinsed with deionized water, bagged, refrigerated, kept dark, and brought back for analysis.

Two sampling regimes were employed: in one method, the resins were deployed for the duration of the study period for nutrient adsorption from June 30 to July 27 (Resin A). In the second method, resins were deployed at the beginning of the study period from June 30 to July 13 (Resin B1) and switched out for new resins halfway through from July 13 to July 27



(Resin B2) to test for the robustness of the resins when comparing the total nutrients adsorbed to the resins deployed for the entire season. This sampling methodology also helps account for the seasonal variation of soil nutrient availability.

## 2.3 Carbon Flux Sampling

NEE and ER measurements were conducted using closed, static chambers, according to methods in Beamish *et al.* (2014). A PVC collar (20 cm diameter) was inserted into the ground (at roughly 5 cm depth) at half of the resin sampling sites and transparent chambers were attached to the collars using a rubber gasket to create a seal. Instantaneous $CO_2$ concentration was measured using a portable infrared gas analyzer (Vaisala GMP343 Carbon Dioxide Probe; $\pm$ 3ppm) (Vaisala, Vaanta, Finland) in sites adjacent to the resin sampling area. A relative humidity and temperature probe (Vaisala HMP75 Relative Humidity and Temperature Probe; $\pm1\%$ RH, $\pm0.2°C$) in the chamber measured these parameters simultaneously. Changes to $CO_2$ concentrations were measured in the chamber at five second intervals for five minutes. After the light measurement, the chambers were removed from the collars and ventilated to ambient conditions, after which an opaque shroud was used to cover the chamber and prevent photosynthesis for the ER measurement. A Kestrel 3500 weather meter was used to determine atmospheric pressure which was needed for the gas flux calculations (Kestrel, Birmingham, Michigan, United States).

## 2.4 Environmental Measurements

Environmental measurements (soil moisture, soil temperature, and active layer depth) were taken twice per week. Sampling sites were adjacent to the locations of the ion exchange resins. Soil moisture (SM) was measured at 0-5 cm depth using a ML-3 Theta probe with data stored in a data logger. Soil temperature (ST) was measured at 5-7 cm depths using a standard soil temperature probe. Active layer (AL) depth was measured using a steel rod that was inserted into the ground until reaching frozen ground. A local meteorological station set up by Queen's University, "West Met" provided hourly temperature and precipitation data.

## 2.5 Post-Field Processing

The ion exchange resins were eluted in groups of 10 strips (5 per cation and anion) creating an integrated "result" for each plot; consequently, single values for each ion of interest were generated for each of the 64 sampling areas. The concurrent elutions to produce one sample for analysis was done to avoid nutrient readings below the detection limit, as Arctic environments are nutrient-limited to begin with. The eluates were extracted by soaking the resins in a 400 mL 0.5M HCl solution for one hour with 40 rotations per minute (RPM) agitation. The elution for each sampling site were down-sampled into two sets of 50 mL centrifuge tubes for transportation and storage prior to colourimetric analysis.

The Astoria2 Analyzer automated colourimetric system analyzed for $NH_4^+$ and $NO_3^-$. The phenolate method was used to determine $NH_4^+$ concentrations and the cadmium reduction method was employed for determining $NO_3^-$ concentrations (Pansu and Gautheyrou, 2006). The samples were 4 mL cuvettes to be run for analysis and three quality





assurance and quality control steps were taken to assess for error: 1 in 10 samples was run in duplicate, 1 in 30 samples was a blank (MilliQ), and 1 in 45 samples was run using blanks to account for instrumentation drift. Overall, all samples were run twice and averaged where valid.

$CO_2$ fluxes and gross primary production (GPP) were calculated using a Matlab script from the $CO_2$ concentration data gathered during each sampling period. $CO_2$ concentrations were converted to fluxes by first converting ppm values to moles of $CO_2$ using the ideal gas law and temperature and pressure data collected during field data acquisition. Gas concentrations ($\mu$mol m$^{-3}$ s$^{-1}$) were calculated from the inputted air pressure (hPa), relative humidity (%), temperature (°C), and gas concentrations in parts per million (Eq. 1):

$$n = C(\frac{\rho}{R\tau})(\frac{v}{A}) \ , \qquad (1)$$

where $n$: converted gas concentration ($\mu$mol m$^{-3}$), $C$ is the original measured gas concentration (ppm), $\rho$ is ambient air pressure (hPa), $R$ is the ideal gas constant (8.314 J K$^{-1}$mol$^{-1}$), $\tau$ is air temperature (K), $v$ is the combined volume of chamber attached to the collar minus the volume of the sensors (m$^3$), and $A$ is the projected horizontal surface area of the chamber (m$^2$). Once converted to moles, an iterative linear regression algorithm went through the inputted data to determine the rate of change during the 5-minute measurement period. The algorithm searches through the data and finds the best subset of points that yields the highest $R^2$ value (minimum of 12 points). The slopes of these regression lines represent the flux rate ($\mu$mol m$^{-2}$). GPP (total photosynthesis) was calculated as the difference between NEE (e.g. overall $CO_2$ exchange) and ER (e.g. $CO_2$ loss through respiration into the atmosphere).

## 2.6 Data Analysis

The IBM SPSS 22 statistical analysis package was used for all statistical tests. Repeated measures ANOVA was used to compare environmental variables, C fluxes, and available soil N in the different moisture regimes. Normality (Shapiro-Wilk's), homogeneity, and sphericity (Greenhouse-Geisser correction) was tested and outliers were removed. A second two-way repeated measures ANOVA explored spatial differences within a single moisture regime (e.g. dry vs. dry, wet vs. wet). For these analyses, Tukey's post-hoc test was run and individual tracks were compared temporally to determine the areas of greatest variance. Bivariate and multiple regression was run with available N as the dependent variable to determine relationships between environmental variables with early, late, and total season $NO_3^-$ and $NH_4^+$. N was used as the independent variable when analyzed against early, late, and total season GPP and ER. The Durbin-Watson statistic determined that there was independence of residuals for all resin regimes. The data were checked for homoscedasticity, multicollinearity, and normality and outliers were removed prior to analysis.

Local indicators of spatial association (LISA) maps were created in GeoDa and analyzed to determine clusters of areas with high and low $NO_3^-$ and $NH_4^+$ tended to gather within Muskox. Ordinary Kriging in ArcGIS 10.3.1 was conducted on total, early, and late season $NO_3^-$ and $NH_4^+$ concentrations to interpolate concentrations across the wet sedge meadow and confirm the spatial autocorrelation results.





# 3 Results

## 3.1 Study Period Air Temperature and Precipitation

The 2016 growing season exhibited similar temperature and precipitation patterns to those for 2014. When comparing the last three growing seasons, 2016 exhibited significantly higher cumulative rainfall and warmer June temperatures, while 2015 had the highest June temperatures of the three years. Maximum air temperature (AT) of 17.4°C was reached on July 5, 2016, which was preceded by precipitation of 2.2 mm the day before. The early season was characterized by little to no precipitation with constant temperature increases, while the later part of the season had more occurrences of precipitation with decreasing AT.

## 3.2 Environmental Variables

Soil temperature was measured ten times during the growing season, and was significantly higher in the wet tracks (1.5 vs. 3.2°C for dry vs. wet, respectively), $F_{(1,28)} = 19.8$, $p < 0.05$ (Table 1). There was also a statistically significant time effect across the four-week growing season, $F_{(1.77,49.4)} = 97.5$, $p < 0.05$, $\varepsilon = 0.59$, with the wetter track means increasing from Week 1 to Week 4 (Figure 2a). However, there was no statistically significant three-way interaction between ST within the moisture tracks over the growing season, $F_{(1.765,49.418)} = 1.281$, $p = 0.284$. When the tracks were compared within a moisture regime (to test spatial variability), the dry tracks differed significantly for within-subjects effects, $F_{(3,33)} = 106.817$, $p < 0.05$, but there was not statistical difference for between-subjects effect across individual dry tracks across the season, $F_{(9,33)} = 1.51$, $p = 0.186$. In the wet tracks, the measurements were statistically significant across weeks, $F_{(3,33)} = 106.1$, $p < 0.05$, and between individual wet tracks across the growing season, $F_{(9,33)} = 10.1$, $p < 0.05$. The post-hoc results indicated the greatest difference between Wet 1 and Wet 5, which was statistically significant ($p < 0.05$).

SM differed significantly between wet and dry tracks, $F_{(1,28)} = 39.4$, $p < 0.001$ (Table 1, Figure 2b). There was a statistically significant time effect between SM across weeks over the growing season, $F_{(2.10,58.7)} = 26.1$, $p < 0.05$, $\varepsilon = 0.70$; as with ST, SM also steadily increased from week to week and measurements were greater in the wet tracks than the dry tracks. When the tracks were compared within moisture regimes, the dry tracks differed between weeks, $F_{(1.65,18.2)} = 24.4$, $p < 0.05$, but not across the season, $F_{(4.96,18.2)} = 1.33$, $p = 0.297$. In the wet tracks, the measurements were statistically significant across weeks, $F_{(1.73,19.1)} = 10.8$, $p < 0.05$, $\varepsilon = 0.551$, but not between individual wet tracks across the growing season, $F_{(5.20,19.1)} = 2.54$, $p = 0.062$, $\varepsilon = 0.551$. The post-hoc results indicated that wet and dry tracks exhibited the greatest within track differences in the western portion of the plot, while the eastern tracks were more similar to each other.

Between-subject effects showed that active layer depth was higher in the wet compared to the dry tracks, $F_{(1,28)} = 24.0$, $p < 0.05$ (Table 1). There was a statistically significant within-subjects effect between AL between weeks throughout the growing season, $F_{(2.31,64.6)} = 74.0$, $p < 0.05$, $\varepsilon = 0.769$. In both the wet and dry tracks, the AL increased from week to week, with the biggest increased in AL happening in the first half of the growing season (Figure 2c). A statistically

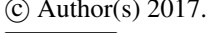



significant interaction was found between AL across the moisture track-week treatment, $F_{(2.31, 64.6)} = 8.36$, $p < 0.05$. When the tracks were investigated individually against each other, the dry tracks were still statistically significant across weeks, $F_{(3,33)} = 17.0$, $p < 0.05$, but there was not statistical difference between individual dry tracks across the season, $F_{(9,33)} = 1.67$, $p = 0.137$. In the wet tracks, the measurements were statistically significant across weeks, $F_{(3,33)} = 74.1$, $p <$

$0.05$, but not between individual wet tracks across the growing season, $F_{(9,33)} = 0.736$, $p = 0.673$. The post-hoc results indicated the greatest difference between Wet 1 and Wet 5 ($p < 0.05$) as well as Wet 1 and Wet 7 ($p = 0.067$). As with SM, the post-hoc results indicated that the greatest within-track differences were in the western portion of Muskox for both wet and dry tracks.

### 3.3 Carbon Flux

GPP was higher in the wet tracks compared to the dry tracks, $F_{(1,22)} = 391$, $p < 0.05$ (Figure 3a). There was also a statistically significant within-subjects effect of GPP across the four-week growing season, $F_{(3,84)} = 8.35$, $p < 0.05$. A significant interaction existed between GPP across the moisture tracks over the growing season, $F_{(1,28)} = 24.0$, $p < 0.05$. As with ER, incoming GPP was greater in the wet tracks than the dry tracks. When the tracks were investigated individually against each other, the within-subjects effects of dry tracks were still statistically different across weeks, $F_{(1,12)} = 51.8$, $p <$

$0.05$, but not between individual dry tracks across the season, $F_{(3,12)} = 1.99$ $p = 0.169$ (Table 2). Examination of the Tukey's post-hoc test results showed the greatest difference in GPP was between Dry 2 and 4, although these differences were not statistically significant. In the wet tracks, the within-subjects effects of wet track measurements were statistically different across weeks, $F_{(1,12)} = 514$, $p < 0.05$, as well as between individual wet tracks across the growing season, $F_{(3,12)} = 11.6$, $p < 0.05$. The post-hoc results indicated the greatest difference between Wet 1 and Wet 5 ($p < 0.05$). Like all other

environmental variables, this greatest difference was located in the western portion of the plot.

ER was significantly higher in the wet tracks, $F_{(1,28)} = 5.46$, $p < 0.05$ (Figure 3b). There was also a statistically significant within-subjects effect between ER across the four-week growing season, $F_{(1.42,39.6)} = 58.5$, $p < 0.05$, $\varepsilon = 0.472$, and a statistically significant interaction between ER across the moisture tracks over the growing season, $F_{(1.42,39.6)} = 6.70$, $p < 0.05$. When the tracks were investigated individually against each other, the within-subjects effects of dry tracks

were still statistically significant across weeks, $F_{(1.79,19.6)} = 47.4$, $p < 0.05$, as well as between individual dry tracks across the season, $F_{(5.35,19.6)} = 3.801$, $p < 0.05$. Examination of the Tukey's post-hoc test results showed the greatest difference in ER was between Dry 2 and 4 ($p < 0.05$). In the wet tracks, the within-subjects effects were statistically significant across weeks, $F_{(2.00,22.0)} = 78.142$, $p < 0.05$, and between individual wet tracks across the growing season, $F_{(6.01,22.0)} = 8.46$, $p < 0.05$. The post-hoc results indicated the greatest difference between Wet 1 and Wet 5 ($p < 0.05$) as well as Wet 1 and Wet

7 ($p < 0.05$).

NEE differed significantly between tracks, being greatest (most negative) in the wet tracks, $F_{(1,28)} = 29.8$, $p < 0.05$) (Figure 4c). Dry tracks were C sources in the early season (atmospheric C gain), transitioning into a late season sink, while wet tracks were net C sinks throughout the entire season (terrestrial C uptake). NEE demonstrated a significant within-





subjects effect across the growing season, $F_{(2.41, 67.6)} = 46.6$, $p < 0.05$, $\varepsilon = 0.804$, but no statistically significant interaction between-subjects effect of NEE across the moisture tracks over the growing season, $F_{(2.41, 67.6)} = 0.847$, $p = 0.452$. The wet tracks were also a net sink, while dry tracks shifted from being an early season source to late season sink, as shown in Figure 3c. When the tracks were investigated individually against each other, the within-subjects effects of dry tracks were

still statistically significant across weeks, $F_{(3,33)} = 29.4$, $p < 0.05$, but there was no statistical between-subject effect between individual dry tracks across the season, $F_{(9,33)} = 1.46$, $p = 0.203$. In the wet tracks, results suggest greatest differences in the western end of the plot. In the wet tracks, the within-subject effects were statistically different across weeks, $F_{(3,33)} = 23.9$, $p < 0.05$, but not between individual wet tracks across the growing season, $F_{(9,33)} = 2.02$, $p = 0.068$. As with SM and AL, the greatest differences were in the western portion of Muskox.

**3.4 Relationships Between Carbon Flux and Environmental Variables**

Pearson's bivariate correlation coefficient was used to explore relationships between environmental variables and C fluxes in the wet sedge meadow. Total season environmental variables showed statistically significant relationships across all measurements (Table 3). Strong positive relationships existed between AL and ST as well as NEE and GPP, while strong negative relationships existed between NEE, GPP, and all environmental variables. As with the entire growing season, AL

and ST were also strongly related to one another in the early season. Early season GPP had strong significant negative relationships with all environmental variables and carbon flux measurements (Table 4). Late season environmental variables and C fluxes were similar to that observed in the early part of the growing season, with AL and ST having the strongest relationship (Table 5). NEE and GPP also correlated positively.

**3.5 Available Soil Nitrogen**

**3.5.1 Seasonal Trends**

There was a statistically significant difference between early and late season $NH_4^+$ availability, $F_{(1,27)} = 98.462$, $p < 0.05$. However, the interaction between seasonality and moisture track was statistically insignificant, $F_{(1,27)} = 0.163$, $p = 0.689$. Overall, $NH_4^+$ availability was greater in the late season than in the early season. The sum of early and late season

$NH_4^+$ adsorption as determined through the ion exchange resins was greater than that of total season resin adsorption, suggesting a potential saturation level reached in the resins left out for the full season.

Nitrate availability also varied seasonally, $F_{(1,27)} = 264.622$, $p < 0.05$. However, the interaction between seasonality and moisture track was statistically insignificant, $F_{(1,27)} = 0.491$, $p = 0.490$. Overall, mean values of $NO_3^-$ in the tracks decreased from early to late season (Resin B2 > Resin B1). The sum of early and late season available $NO_3^-$ was

greater than that of total season resin adsorption, suggesting a potential saturation of the resins and a need for further investigation.



### 3.5.2 Spatial Patterns

Local spatial autocorrelations for N were run and the local indicators of spatial association (LISA) maps were analyzed to determine where areas of high and low $NO_3^-$ and $NH_4^+$ tended to gather within Muskox. High-high values of N ($p < 0.05$) tended to gather in the southwestern portion of the plot in all resins except late season $NH_4^+$, while low-low values

of N ($p < 0.05$) clustered in the northeastern corner of the plot for all resin regimes of $NO_3^-$ and $NH_4^+$. An examination of interpolation maps created through Kriging analysis in ArcGIS showed similar trends in $NO_3^-$ and $NH_4^+$ (Figure 4), with higher concentrations of the nutrients in the southwestern corner of the plot and lowest values in the northeastern portion. This pattern followed the gentle sloping of topography in the meadow, where the northeast corner was at the highest elevation, sloping down to the southwest corner at the lowest point.

The early season trend of $NH_4^+$ was similar to that of total season $NH_4^+$, with the highest values being in the western portion of the plot. However, early season $NH_4^+$ is located more north than the end location of total season $NH_4^+$. No clear patterns are discernable from the early season $NO_3^-$ kriging map, aside from the location of highest values being in the southeastern corner of the plot. When looking at late season N, the $NO_3^-$ and $NH_4^+$ patterns almost seem like mirror images to those of the early season; late season $NH_4^+$ matched early season $NO_3^-$ patterns, while late season $NO_3^-$ matched early

season $NH_4^+$ patterns. As compared to the early season, late season $NO_3^-$ had shifted from the southeastern corner to the southwestern corner of the plot. Late season $NH_4^+$ was highest in the southeastern corner and lower on the western side of the plot.

### 3.6 Relationships Between Available Nitrogen, Carbon Fluxes, and Environmental Variables

Linear regression analysis showed that across the dry tracks, AL and total season $NO_3^-$ ($R^2 = 0.707$), $F(1,12) =$

28.937, $p < 0.05$, and ST and total season $NO_3^-$ ($R^2 = 0.598$), $F(1,12) = 17.840$, $p < 0.05$ were significantly related (Table 6). Total season $NH_4^+$ in the dry tracks were significant but moderate in their relationships with AL ($R^2 = 0.470$), $F(1,12) = 10.633$, $p < 0.05$, and ST ($R^2 = 0.456$), $F(1,12) = 10.046$, $p < 0.05$. Carbon flux measurements also showed significant but moderate relationships between total and late season $NO_3^-$ and GPP ($r^2 = 0.323$ and $R^2 = 0.338$), $F(1,14) = 6.64$, $p < 0.05$ and $F(1,14) = 6.20$, $p < 0.05$ respectively, as well as moderate relationships with between total season $NH_4^+$ and ER ($R^2 = 0.359$)

and GPP ($R^2 = 0.348$), $F(1,14) = 7.27$, $p < 0.05$ and $F(1,14) = 6.93$, $p < 0.05$ (Table 7). Early season $NO_3^-$ and early and late season $NH_4^+$ were not significant or strong in their relationships to biophysical variables and carbon fluxes, suggesting that in a shorter time scale, soil available nitrogen is not a strong predictor of potential ER or GPP.

In the wet tracks, total season SM had a strong and significant relationship with $NO_3^-$ ($R^2 = 0.646$), $F(1,13) =$ 23.765, $p < 0.05$ (Table 8). Total season $NH_4^+$ had significant relationships with AL and ST as well. ER and GPP were

moderate predictors strong predictors of total season available soil $NO_3^-$, while early season $NH_4^+$ predicted ER (Table 9). Early season $NO_3^-$ and late season $NH_4^+$ were not significant or strong in their relationships to biophysical variables and carbon fluxes.



Multiple regression models were explored to predict C fluxes using both soil available N variables and environmental variables. Three of the models were strongly and significantly predicted for using AL, ST, SM, $NO_3^-$, and $NH_4^+$: total season GPP in the wet tracks ($R^2 = 0.685$) and early season GPP in both dry ($R^2 = 0.788$) and wet tracks ($R^2 = 0.785$) (Table 10). None of the models predicting ER were significant at $p < 0.05$.

## 4 Discussion

### 4.1 Carbon Flux and Environmental Trends

As expected with the progression of the growing season, ST and SM values increased while AL deepened ($p < 0.05$); this matched results of environmental trends from previous studies by Blaser and Luce in Cape Bounty wet sedge plant communities (Blaser, 2016; Luce, 2016). The local spatial differences of wet and dry tracks for C exchange and environmental variables were all significant; generally, wet tracks had warmer, wetter, and deeper values than dry tracks for ST, SM, and AL. Perhaps the most notable difference between SM in the wet and dry tracks is the consistent increase of SM across the season, while there is a decrease in SM in dry tracks between weeks 3 and 4. This finding is an indication of the start of drying out and return to fall conditions. This difference between the moisture tracks indicative of the need to account for fine-scale variability when characterizing wet sedge communities across the landscape.

Examinations of 2014 and 2015 C fluxes in the wet sedge determined that these wetland communities were a net C sink (Blaser, 2016; Luce, 2016). In examining 2016 C flux data, the wet sedge meadow was an early season source with ER dominant with a net efflux of C, turning into a late season sink as GPP took over with a net C influx. There was spatial variability between the moisture regimes of the wet sedge meadow, with GPP and ER higher in wet tracks than dry tracks, contrary to previous results where dry sites yielded higher cumulative GPP (Blaser, 2016). Over the growing season, dry tracks averaged 1.513 µmol/m$^2$/s in ER while mean ER in the wet tracks was 1.899 µmol/m$^2$/s. GPP calculated using NEE and ER measurements averaged -1.34 µmol/m$^2$/s and -3.01 µmol/m$^2$/s in the dry and wet tracks respectively. Overall, dry tracks yielded a 0.171 µmol/m$^2$/s seasonal NEE average, while wet tracks had an NEE seasonal average of -1.112 µmol/m$^2$/s. These numbers are indicative of the Muskox wet sedge meadow being an overall sink in the 2016 growing season, and suggest an important role for soil moisture as a limiting factor for plant growth in this environment. ER was the dominant process for C fluxes in the early season dry tracks, switching over to GPP being the dominant process late in the season. However, GPP was the dominant factor throughout the entire season in the wet tracks, suggesting the importance of the role that SM has in photosynthetic processes in the High Arctic wetlands.

Across the entire season, increases in ST strongly corresponded with a deepening active layer. All other variables were also moderately related, except for SM and AL. Early season GPP and NEE had moderate negative relationships with AL, ST, and SM, while increased ST and SM and deeper AL corresponded to higher ER. ST increases across the growing season corresponded a net C release with decreasing NEE. In the early season, ST governed ER, but the transition into the late season saw a shift of ST influencing GPP. Slowing rates of ER in the latter parts of the season corresponded with



deepening active layers and higher ST, evidence of increased decomposition with increased temperatures and lowering water table as shown in literature (Schuur *et al.*, 2009; Guicharnaud *et al.*, 2010). As the season progressed all C flux relationships with AL, ST, and SM became negative. Wet and dry tracks exhibited the similar temporal patterns.

### 4.2 Nitrogen Trends

$NO_3^-$ and $NH_4^+$ were significantly different across wet and dry tracks, again reinforcing the importance of fine-scale spatial variability that can exist within a generally homogenous ecosystem. Across most tracks, it was determined that wet tracks generally had higher levels of $NH_4^+$ than dry tracks, while $NO_3^-$ concentrations were similar across both moisture gradients. Raw concentrations of $NH_4^+$ were always higher than concentrations of $NO_3^-$, which is to be expected because of their relative positions and roles in the N cycle and is in accordance with results from prior research (Nadelhoffer *et al.*,

1991; Clein and Schimel, 1995; Gregory, 2011). Furthermore, in an anoxic environment like Arctic wetlands, aerobic processes like nitrification are generally limited or absent (Giblin *et al.*, 1991; Nadelhoffer *et al.*, 1991; Clein and Schimel, 1995; Stark, 2007; Beermann, 2016). On average, N adsorption was higher in the later season (July 13 to 27) than the earlier part of the season (June 30 to July 13). This is contrary to findings from 2008 wet sedge nutrient measurements where early season adsorption was higher than the late season (Gregory, 2011); however, the prior study only deployed one PRS

measurement in each wet sedge site (for a total of 4), as such the spatial variability is not accounted for. Compared to the polar desert and mesic tundra N PRS probe measurements from 2008 (Gregory, 2011), the values measured in the wet sedge meadow are generally two to three times higher. The later season nutrient release and subsequent adsorption can be explained by a combination of early season snowmelt, increasing ST, and a deepening active layer that allowed for water flow, promoting nutrient availability and the release of inorganic N. A lag period follows late in the early season after which

organic N is then available for microbes to mineralize into $NH_4^+$ and then $NO_3^-$.

The bivariate regression statistics showed that AL and ST have strong positive relationships with $NO_3^-$ and $NH_4^+$ in the dry tracks. The seasonal $NO_3^-$ and $NH_4^+$ measurements could not be predicted significantly or strongly by a single variable. SM in the wet tracks was able to predict total and late season $NO_3^-$, which could be indicative of the role of water in nitrification processes. Multivariate regression analyses showed much stronger relationships between environmental

variables and $NO_3^-$ and $NH_4^+$ concentrations in predicting GPP, which indicates that multiple factors influence photosynthetic activity in wet sedge. While dry track models were not significant, the multivariate regression models favoured wet tracks, strengthening the argument of the importance of water in the processes in wet sedge meadows. In the wet tracks, the multivariate regression model suggests that total season GPP is most affected by ST and $NO_3^-$. Increases in $NO_3^-$ would also have a greater effect on GPP decrease than $NH_4^+$ would. Early season GPP is most affected by ST, and the

soil available N impacts on GPP are similar.

The results from the regression models point to increases in ST having the greatest effect on mineralization and nitrification as compared to other environmental variables. Furthermore, in the dry tracks, the correlation with ST was greater for nitrification than mineralization, while the opposite effect was seen in the wet tracks. This was consistent with





results from tundra studies in Toolik Lake, Alaska which found that experimentally elevated temperatures triggered increased N mineralization, while light attenuation resulted in decreased $NH_4^+$ levels (Shaver and Chapin., 1995). Subsequent experiments with temperature-driven greenhouse incubations found consistent increases in N mineralization (Shaver *et al.*, 1998) at magnitudes of 40-200%. At CBAWO, increases in SM promoted nitrification towards the latter part

of the season, as reflected in the higher late and total season $NO_3^-$. Nitrification was always secondary to mineralization and absent in some plots; this is a change from previous research where nitrification was often completely absent in wet sedge tundra (Giblin *et al.*, 1991; Stark, 2007).

The southern portion of the meadow generally had higher $NO_3^-$ and $NH_4^+$ than the north and the west had higher N than the east; this can be attributed to the downslope of the topography promoting water flow, which is a transport method of

nutrients down the meadow into lower lying areas (Woo and Young, 2006). This is consistent with research conducted by Stewart *et al.* (2014), where N mineralization rates, and consequently $NH_4^+$ levels, were higher in lower lying areas. The increase of nutrient concentrations from east to west also follows the east to west retreat of the snowpack as the melt season progressed, which ties back to the importance of this perennial snowpack to the wetland in this environment (Woo and Young, 2014).

**4.2 Comparing the Influence of Abiotic Conditions and Nitrogen Availability on Carbon Exchange**

The relative importance of nutrients in contributing to GPP and ER is often overlooked. Many studies in northern regions examine environmental variables as the main contributors to influencing carbon dynamics in these ecosystems; however, this study finds nitrate as a strong predictor of ER (Figure 5) and GPP (Figure 6), explaining 37.0% and 50.6% of the variation respectively. In an environment not favourable for nitrification to readily occur, the magnitude of the role of

nitrate has in carbon exchange processes warrants further research.

Early spring and late fall measurements important pieces to add for a more robust and comprehensive study. While the growing season presents opportunity to investigate the most productive time for these ecosystems, it might not be a good way to study annual carbon fluxes. Research through snow manipulation has shown that previous seasons' winter and spring climatic events play a role in summer and year-long growth (Robinson, 2002; Aerts *et al.*, 2005; Edwards *et al.*, 2006).

Edwards *et al.* (2006) found peak nutrient availability to be early in the freeze-thaw period when soil temperatures were between -7 and 0°C; as such, solely measuring growing season nutrient availability does not encompass the period when soil-available nutrients may be at their peak. Furthermore, research has found microbial nutrient cycling to be present in Arctic snowpacks (Larose *et al*, 2013), and the release of these snowpack nutrients during melt can affect vegetation cover and productivity in the following growing season. Understanding the role that winter freeze and spring thaw has on Arctic

ecosystems can help determine and project future shifts in plant cover and soil composition that are anticipated results of climate change (Aerts *et al.*¸2005).

Historically, Arctic wetlands have been a strong C sink (Mikan *et al.*, 2005), and with projected increases in air temperature affecting soil temperature, increases in soil respiration have the possibility of creating direct positive feedback



loops (Post *et al.*, 1982; Elberling *et al.*, 2008; Chae *et al.*, 2015; Christiansen, 2016; Euskirchen *et al.*, 2016). In this study, the 2016 growing season saw the wetland plot shift from being an early season source to a late season sink. Wet tracks were generally sinks throughout the entire season, while the dry tracks saw the shift from source to sink, and again the magnitude of the flux processes in wet tracks were always greater than in the dry tracks. This conforms with research from Welker *et al.*

(2004) where the differences between land cover C exchange responses had a strong dependence on hydrologic conditions and that wet sedge productivity is strongly linked to moisture (Reynolds and Tenhunen, 1996). The predicted future increases in air temperature will promote earlier and deeper thaw, allowing for microbial activity to be active and for movement and transfer of nutrients (Biederbeck and Campbell, 1973; Jonasson and Shaver, 1999; Shaver *et al.*, 2000). The spatial and temporal dynamics of these nutrients will dictate future shifts in biotic and abiotic conditions.

The distribution and role of N-fixing cyanobacteria needs to be investigated in tandem with belowground inorganic N to get the full spectrum of High Arctic N cycling in wetlands. Quantifying the N fixation of *Nostoc* using stable isotope and chemical analysis would allow for the determination of the contribution of the cyanobacteria in N cycling in an environment like High Arctic wet sedge (Skrzypek *et al.*, 2015). High $N_2O$ concentrations have been linked to higher $NH_4^+$ levels belowground (Stewart *et al.*, 2014), which could be explored through the analysis of trace gas samples in tandem with

nutrient adsorption (trace gas samples were taken of the 2016 growing season in Muskox but not included in this manuscript).

Improving our understanding of N dynamics within a complex ecosystem like a High Arctic wetland, requires other contributing nutrients with C and phosphorus to be examined for their roles in promoting or limiting different processes within the N cycle. Phosphorus has long been known to be a limiting nutrient in Arctic plant growth (Nadelhoffer *et al.*,

1992; Shaver and Chapin, 1995; Shaver *et al.*, 1998; Shaver *et al.*, 2000; Chapin *et al.*, 2002; Stark, 2007), and in some experiments, has been shown to be the dominant limiting nutrient in Arctic wet sedge (Shaver *et al.*, 1998; Gough and Hobbie, 2003). The balance between N and phosphorus in these environments can have effects on the rates and shifts in microbial activity (Shaver *et al.*, 1998), and future nutrient cycling studies need to incorporate both limiting nutrients in examining Arctic wetlands. The importance of the role of C in High Arctic N cycling has also been presented in literature

(Stark, 2007). C to N ratios have been shown in literature to be an important proxy of mineralization rates (Janssen, 1996), and the ratio of mineralized C to mineralized N is overall affected by soil temperature (Robinson, 2002). Regardless of the environment and temperature, the important interactions between C and N need to be considered in future studies of N cycling.

## 5 Conclusions

The 2016 growing season saw increased ST and AL as the season progressed, resulting in higher $NH_4^+$ and $NO_3^-$ in the latter part of the season. Mineralization was the dominant portion of the N cycle over nitrification, which is consistent with previous Arctic N studies (Giblin *et al.*, 1991; Stark, 2007). Nitrification was present throughout the growing season, as





evidenced by the increasing presence of $NO_3^-$ from the early season to late season. This is different from previous studies where nitrification was generally absent from wet sedge meadows (Giblin *et al.*, 1991; Stark, 2007). SM played a role in nitrification, but only in the latter half of the growing season. With the sampling methodology that was employed, it was qualitatively demonstrated that water availability plays an important role in the regulation of environmental conditions in this

wet sedge meadow. The multivariate regression models had high goodness of fit for all wet tracks, implying that in the presence of higher soil water content, the quantitative relationship between soil N and environmental variables can be better predicted. Over the growing season, the wet sedge meadow was an early season source and late season sink for carbon; overall, the system was a net sink. GPP increases throughout the first three weeks of the growing season corresponded with increases in $NH_4^+$, suggesting that N mineralization may promote photosynthetic activity. Billings *et al.* (1984) found that

that an increase in soil N significantly increased $CO_2$ uptake; this relationship was also found in the wet sedge of CBAWO, as regression models suggested $NH_4^+$ was a driver of GPP. Proximity to the perennial snowpack had an indirect effect on N availability in the High Arctic wetland. Discrete distance to the snowpack had no direct effect, but the snowpack's function as a water source was evident as the melt season progressed. The north to south and east to west downslopes of the wetland acted as mechanisms for water flow from snowmelt, which was a transport method of inorganic N. As such, higher levels of

$NH_4^+$ and $NO_3^-$ were found in lower-lying areas of the wetland. However, further work is needed to examine the role the nutrient-spiraling concept may have played in N distribution of this downsloping wetland.

The study confirms that multiple environmental variables contribute to determining nitrogen concentrations in wet sedge meadows, and soil N availability does not show a straightforward response to increases in temperature (Robinson, 2002). Multivariate regression models of environmental variables versus inorganic N performed significantly better than

bivariate regression models of individual environmental measures against $NO_3^-$ and $NH_4^+$. This is indicative of the important of multiple environmental variables in controlling the nutrient dynamics of a High Arctic wetland; similar results have been found in other High Arctic wet sedge studies (Shaver *et al.*, 1998). The study also found that, ST is a driver of N mineralization, with increased ST coinciding with higher $NH_4^+$ concentrations, promoting GPP. This conforms with previous investigations of drivers of Arctic N cycling that N mineralization is highly temperature dependent (Biederbeck and

Campbell, 1973; Billings *et al.*, 1982; Nadelhoffer *et al.*, 1991; Shaver *et al.*, 1998; Rustad *et al.*, 2001; Robinson, 2002), as the coupling of light and ST as light is a contributing factor to warmth. Increased $NH_4^+$ to plants can be incorporated directly into their $NH_4^+$ assimilation pathways, which will then allow them to harness $CO_2$ and water to produce glucose, which is what gross primary production encompasses. The inorganic nitrogen dynamics cannot be fully analyzed separately from the hydrological dynamics in High Arctic wetlands, particularly when there is an elevation gradient present. The transport and

distribution of inorganic N in this High Arctic wetland is highly dependent on the movement of water, as the wet sedge meadow is located on a downslope. Research has suggested that spatial vegetation patterns are highly dependent on underlying hydrology for the distribution of nutrients necessary for growth (Oberbauer *et al.*, 1989; Rastetter *et al.*, 2004), and as such the nutrient-spiraling models can be applied in these types of terrestrial environments. Using the knowledge gathered from this study and the hillslope-nutrient model developed by Rastetter *et al.* (2004), the movement of N




downslope is a slow process, and downslope additions of plant-available N will result in increased photosynthetic rates (Oberbauer *et al.*, 1989).

The spatial and temporal dynamics of a High Arctic wet sedge meadow exhibit variances within the ecosystem itself and investigations into the processes operating in the wetlands cannot simply be scaled up to the biome level (Rustad *et al.*, 2001). This study found that seasonally, plant-available N was highest in the latter part of the growing season, with the largest concentrations of these inorganic N forms in lower-lying areas. Increases in GPP corresponded to increases in $NH_4^+$, demonstrating a link between light and photosynthetic activity with mineralization rates. Nitrification, although muted, was present in this environment in the 2016 growing season. Within the plot, significant differences were found between moisture tracks: the patterns were the same between the tracks, but the rates at which these patterns occurred were higher in wet tracks than dry tracks. This finding is indicative of the importance of water availability and moisture in driving the abiotic processes that occur in wet sedge meadows; the next step in this finding would be to determine the magnitude of the role water plays in controlling each of these variables. The underlying hydrology and movement of water was a large factor in determining the spatial pattern of $NH_4^+$ and $NO_3^-$ in the meadow. The melt of the north-adjacent perennial snowpack as the growing season progressed was the source of water that controlled C flux processes in the meadow. The downslope nature of the Muskox wet sedge meadow allowed for the formation of subsurface preferential flow pathways, transporting and cycling nutrients through the plot (Rastetter *et al.*, 2004).

## 6 Code Availablility

N/A

## 7 Data Availablility

The data will be available to the public shortly on the Polar Data Catalogue.

## 8 Sample Availablility

N/A

## 9 Appendices

N/A




## 10 Supplement link

N/A

## 11 Team List

J.K.Y. Hung, D.M. Atkinson, N.A. Scott, S.Z. Arruda, R. MacTavish

5 **12 Author Contribution**

J.K.Y. Hung and D.M. Atkinson designed the experiments and J.K.Y. Hung carried them out. D.M. Atkinson developed the script used to calculate carbon flux. J.K.Y. Hung prepared the manuscript with contributions from all co-authors. N.A. Scott contributed to manuscript editing and statistical analysis.

## 13 Competing interests

10 The authors declare that they have no conflict of interest.

## 14 Disclaimer

N/A

## 15 Special issue statement

N/A

15 **16 Acknowledgements**

Operations at the Cape Bounty Arctic Watershed Observatory are run by Dr. Scott Lamoureux of Queen's University. Logistical and equipment support was provided by Polar Continental Shelf Program (Natural Resources Canada). Funding from Ryerson University's Geography and Environmental Studies department, the Yeates School of Graduate Studies, and ArcticNet made this research possible.



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



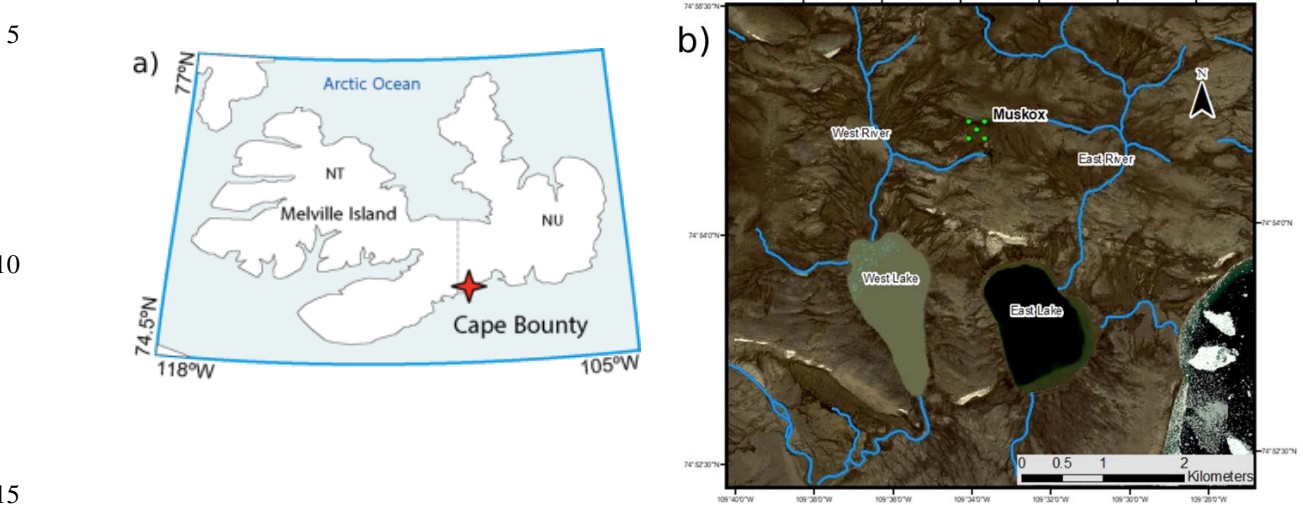

**Fig. 1. (a) Map showing Melville Island with the Cape Bounty Arctic Watershed Observatory (red star). (b) WorldView-2 image of the Cape Bounty Arctic Watershed Observatory, highlighting the study area "Muskox".**

**Table 1. Two-way ANOVA p-values for environmental variables across the growing season in different moisture tracks across the plot. Bolded values indicate significance at $p < 0.05$.**

| Interaction | Soil Temperature (ST) | Soil Moisture (SM) | Active Layer Depth (AL) |
|---|---|---|---|
| Time | **0.000** | **0.000** | **0.000** |
| Moisture track | **0.000** | **0.000** | **0.000** |
| Time x Moisture track | 0.286 | **0.000** | **0.000** |
| Time x Moisture track x Spatial location (W vs. E) | **0.000** | **0.001** | **0.005** |




**Fig. 2. (a) Mean (± 1 SE) ST change across the growing season within the tracks. (b) Mean (± 1 SE) SM change across the growing season. (c) Mean (± 1 SE) AL change across the growing season.**





**Table 2. Two-way ANOVA p-values for carbon flux measurements across the growing season in different moisture tracks across the plot. Bolded values indicate significance at p < 0.05.**

| Interaction | NEE | ER | GPP |
| --- | --- | --- | --- |
| Time | **0.000** | **0.000** | **0.000** |
| Moisture track | **0.000** | **0.027** | **0.000** |
| Time x Moisture track | 0.452 | **0.007** | **0.000** |
| Time x Moisture track x Spatial location (W vs. E) | **0.077** | **0.000** | **0.000** |



**Fig. 3. (a) Mean (± 1 SE) GPP change across the growing season within the tracks. (b) Mean (± 1 SE) ER change across the growing season. (c) Mean (± 1 SE) NEE change across the growing season.**





**Table 3. Pearson's bivariate correlation coefficients for carbon flux measurements and environmental variables across the entire growing season. Bolded values indicate significance at p < 0.05.**

|  | AL | ST | SM | NEE | ER |
|---|---|---|---|---|---|
| ST | **0.963** | | | | |
| SM | **0.558** | **0.552** | | | |
| NEE | **-0.709** | **-0.699** | **-0.700** | | |
| ER | **0.604** | **0.603** | **0.554** | **-0.366** | |
| GPP | **-0.797** | **-0.789** | **-0.769** | **0.923** | **-0.696** |

10   **Table 4. Pearson's bivariate correlation coefficients for carbon flux measurements and environmental variables across the early season. Bolded values indicate significance at p < 0.05.**

|  | AL | ST | SM | NEE | ER |
|---|---|---|---|---|---|
| ST | **0.944** | | | | |
| SM | **0.601** | **0.604** | | | |
| NEE | **-0.634** | **-0.640** | **-0.754** | | |
| ER | **0.647** | **0.655** | **0.512** | -0.231 | |
| GPP | **-0.814** | **-0.823** | **-0.819** | **0.825** | **-0.740** |

**Table 5. Pearson's bivariate correlation coefficients for carbon flux measurements and environmental variables across the late**
20   **season. Bolded values indicate significance at p < 0.05.**

|  | AL | ST | SM | NEE | ER |
|---|---|---|---|---|---|
| ST | **0.966** | | | | |
| SM | **0.464** | **0.453** | | | |
| NEE | **-0.668** | **-0.671** | **-0.542** | | |
| ER | -0.061 | -0.138 | 0.064 | -0.126 | |
| GPP | **-0.660** | **-0.663** | **-0.496** | **0.928** | -0.276 |





**Fig. 4. Ordinary kriging maps for (a) total season adsorption of NO₃⁻ and NH₄⁺, (b) early season adsorption of NO₃⁻ and NH₄⁺, and (c) late season adsorption of NO₃⁻ and NH₄⁺ defined by natural breaks. Red dots represent wet tracks and green dots represent dry tracks.**




**Table 6. Bivariate regression $R^2$ coefficients for nitrogen (dependent variable) against environmental variables (independent variable) across dry tracks. Bolded values indicate significant values at $p < 0.05$.**

|    | Resin A $NO_3^-$ | Resin A $NH_4^+$ | Resin B1 $NO_3^-$ | Resin B1 $NH_4^+$ | Resin B2 $NO_3^-$ | Resin B2 $NH_4^+$ |
|----|----|----|----|----|----|----|
| AL | **0.707** | **0.470** | 0.000 | 0.069 | **0.326** | 0.205 |
| ST | **0.598** | **0.456** | 0.009 | 0.037 | **0.342** | 0.161 |
| SM | 0.026 | 0.021 | 0.011 | 0.150 | 0.013 | 0.017 |

**Table 7. Bivariate regression $R^2$ coefficients for nitrogen (independent variable) against environmental variables (dependent variable) across dry tracks. Bolded values indicate significant values at $p < 0.05$.**

|    | Resin A $NO_3^-$ | Resin A $NH_4^+$ | Resin B1 $NO_3^-$ | Resin B1 $NH_4^+$ | Resin B2 $NO_3^-$ | Resin B2 $NH_4^+$ |
|----|----|----|----|----|----|----|
| ER | 0.209 | **0.359** | 0.002 | 0.024 | 0.087 | 0.100 |
| GPP | **0.323** | **0.348** | 0.097 | 0.095 | **0.338** | 0.060 |

**Table 8. Bivariate regression $R^2$ coefficients for nitrogen against environmental variables and carbon flux across wet tracks. Bolded values indicate significant values at $p < 0.05$.**

|    | Total Season $NO_3^-$ | Total Season $NH_4^+$ | Early Season $NO_3^-$ | Early Season $NH_4^+$ | Late Season $NO_3^-$ | Late Season $NH_4^+$ |
|----|----|----|----|----|----|----|
| AL | 0.227 | **0.370** | 0.134 | 0.219 | 0.127 | 0.013 |
| ST | **0.289** | **0.420** | 0.100 | 0.230 | 0.194 | 0.000 |
| SM | **0.646** | 0.054 | 0.033 | 0.001 | **0.449** | 0.094 |

**Table 9. Bivariate regression $R^2$ coefficients for nitrogen (independent variable) against environmental variables (dependent variable) across wet tracks. Bolded values indicate significant values at $p < 0.05$.**

|    | Total Season $NO_3^-$ | Total Season $NH_4^+$ | Early Season $NO_3^-$ | Early Season $NH_4^+$ | Late Season $NO_3^-$ | Late Season $NH_4^+$ |
|----|----|----|----|----|----|----|
| ER | **0.395** | 0.112 | 0.008 | **0.528** | 0.037 | 0.024 |
| GPP | **0.384** | 0.023 | 0.086 | 0.251 | 0.010 | 0.121 |

**Table 10: Multiple regression $R^2$ values for ER and GPP across wet and dry tracks and seasons. Bolded values indicate significance at $p < 0.05$.**

|    | Dry Tracks | Wet Tracks |
|----|----|----|
| Total season ER | 0.506 | 0.467 |
| Early season ER | 0.426 | 0.642 |
| Late season ER | 0.224 | 0.515 |
| Total season GPP | 0.623 | **0.685** |
| Early season GPP | **0.788** | **0.785** |
| Late season GPP | 0.528 | 0.220 |







**Fig. 5. Ability of environmental variables and inorganic nitrogen in predicting ecosystem respiration.**



**Fig. 6. Ability of environmental variables and inorganic nitrogen in predicting gross primary productivity.**