# Peer review of "Controls on spatial and temporal patterns of soil nitrogen availability in a High Arctic wetland"

_Biogeosciences, 2017_

## Referee Comment (RC1) · Anonymous Referee #1 · 18 Dec 2017

General comments:

Overall, this paper provides a clear method section and presents an extensive analysis of a dataset of available inorganic nitrogen in a high arctic wet sedge meadow. The spatial extent of data collection is welcome and given the importance of arctic wetlands, this study could provide a valuable addition from a different site. There is definitely value in improving understanding the controls on C and N cycling in arctic wetlands and there are potentially interesting data here. The manuscript, however, would benefit from an appropriate title, clearer aims and a greater effort to highlight what novel contribution the study makes. There is a substantial quantity of introductory

text and results, and given that a fairly high proportion of the material confirms existing knowledge (such as the importance of hydrology in determining abiotic processes in wetlands, the importance of fine scale variability in Arctic ecosystems, the multiple environmental drivers of soil nutrient availability), a more concise summary of previous work would leave more room to explore novel findings. In view of the fact that the main issues are mostly to with approach to structure and content, I have provided comments in combination with the minor technical notes.

Specific and technical comments:

Title and abstract - the abstract has a strong focus on N availability and fits well with the title, but a substantial proportion of the results and discussion relate to C exchange. Title and abstract should be modified to better reflect the content of the rest of the paper.

The introduction would benefit greatly from a reduction in length, through taking a more focused approach to presenting the background information and rationale specific to this particular study. The current information is a sort of mini-review demonstrating the breadth of understanding of various elements of arctic ecosystem function but the text moves rapidly between wider issues such as global climate change (p3 lines 3-7), and the specifics of the study several times and this makes it challenging to disentangle exactly what the current understanding is and what gaps this study addresses. Specific sections - p2 line 16 to p4 line 6 includes multiple statements where the relevance isn't clear (for example, the lack of long-term studies in the Arctic, when this a short term study), and p3 lines 20-30 could be summarized in much less text. The sentence at lines 4-6 is one of the few mentions of the key information underpinning this study and yet it is not stated what the past research is or what it showed.

P2, line 15 – 'this study looked at spatial patterns to see how these patterns shift through the growing season' – I presume what is meant is that spatial variability and temporal variability were investigated (doesn't make sense).

P2, line 15 and elsewhere – the investigation of growing season variability seems over-stated – is two sampling periods sufficient to investigate temporal patterns of nutrient availability, even in such a short season?

P2, line 28 "global C changes" is vague – clarify what processes are being referred to here.

P4, line 5 – seems to be the first mention of the CBAWO wetlands, if so, define abbreviation.

P4, line 7 – not sure 'nutrient regime in plant growth' is the right phraseology for what is meant here.

P4, line 25 – this sentence is unclear – what is meant by 'help in future predictions of decreasing carbon storage?'

P5, line 11 – what is a 'spatial lens'? Please clarify what exactly it is that hasn't been previously investigated.

Methods

Throughout methods paper refers to early and late season. This implies that the experiment took place either side of the 'mid' season, when actually it's a main growing season experiment that doesn't include 'early' and 'late' growing season (as acknowledged by the authors in the second paragraph of section 4.2). Suggest either using real time descriptions or clarifying in the methods section (2.2) where the sampled timed periods fit in the overall growing season.

P6, lines 4 and 5 – it is not clear how the vegetation communities differ between wet and dry tracks (i.e. do sedges and grasses dominate, with lesser elements of Salix arctica and herbaceous flowering plants in each of the different types, or does Salix arctica dominate in wet tracks?) With the genera (Carex and Eriophorum), can the authors say whether they're referring to two or more species (spp.) or one unspecified species (sp.)? Also, is there a word (herbaceous?) missing between flowering and

'Eriophorum' (and if so, is this appropriate description as Eriophorum belongs to the Cyperaceae). I think Gramineae is now Poaceae. I appreciate that biotic controls are outside of the scope of this experiment, but there is a surprising lack of mention of the possible role of vegetation in any of the relationships described later, given that different species / plant functional types preferentially use different N sources (e.g. NH4, NO3, organic N).

P6, line 13 – alternate? (not alternative)

P6, line 15 – was? (not is)

Results – is there a reason why this is structured in a different order from the methods? Consistency would make for an easier read.

P9, lines 3-5 – the comparison of data from this study with 2014 and 2015 comes out of nowhere – what is the importance of the June 2015 temperature to this study?

P9 line 19, and P10, lines 5-8 and line 15 – where are the different tracks and what general concept does comparisons of difference between them relate to? This issue is common throughout much of the results (the spatial findings in particular) - references to what is happening on specific parts of the site by name rather than in context don't indicate to the reader how the findings of this study can be applied beyond this specific site.

P10, line 13 – incoming GPP?

P11, line 11 and P12, line 2 – opening sentences should be in the methods, not results

P11, line 25 – this is a common finding with ion exchange resins in arctic ecosystems – saturation seems improbable when deploying the same resins in more nutrient rich ecosystems records often much higher totals, so could it be that in a longer burial some kind of equilibrium with soil levels is reached?

P11, line 30 – further investigation of what?

Discussion and conclusions

These should be checked to see whether the data provide evidence to support all the assertions made, especially where they relate to processes that were assumed rather than measured - such as mineralization, nitrification and transport.

Although there is discussion is context of other studies, the discussion is lacking in implications and does not clearly demonstrate what it is that this study shows that does not simply agree with previous findings (the majority). Where differences are highlighted (e.g. the presence of nitrate) no further thoughts are provided as to why this might be or what this will mean.

Substantial parts of section 4.2 are suggestions for other studies – although this is interesting, it doesn't relate to the sub-heading and it could be summarized in a couple of sentences, rather than providing a heavily referenced rationale.

In the conclusions, some of them seem not arise from the results presented (for example, was there a test of the relationship between distance from snowpack on N availability, and is there any evidence that mineralization promotes photosynthetic activity?) and many of them are readily referenced to other older studies that it is not clear what has been found that is not already well known.

Figs 5 and 6 – add y-axis labels.

Table 7 – title doesn't match table contents.

---

## Referee Comment (RC2) · Anonymous Referee #2 · 25 Dec 2017

Hung et al. examined spatial heterogeneity in soil nutrient pools, effects of prospective abiotic drivers of nutrient availability, and relationships of nutrients and soil moisture with carbon balance of a High Arctic ecosystem. The study identifies landscape positions and times within the growing season that support strong links between nutrients and productivity. Empirical studies such as this have potential to reveal relationships between source-sink dynamics of carbon and spatial and temporal variation in soil moisture that have previously been unrecognized. This study quantifies correlational relationships among carbon fluxes, nutrient availability, and abiotic attributes of soils and extends these correlations to assess mechanistic relationships. Greater caution and scrutiny must be applied to interpreting correlational relationships to consider al-

ternative explanations that do not include direct causal links between the measured attributes. For example, coherent temporal patterns between nitrogen pools and productivity might result if both are responding to a shared driver, and might not reflect a direct effect of nitrogen on plant production. The manuscript's context is broad relative to the limited spatial and temporal extent of data collection. There is value in such focused studies, as they can reveal key patterns that might affect processes at larger scales (e.g., regional C balance), but the patterns revealed by the current analyses and their potential implications tend to get lost among discussions of tangential processes not directly addressed by the data in-hand (e.g., phosphorus, N transformations). Finally, there is a missed opportunity to compare patterns in soil nutrients with nitrogen dynamics at the watershed scale, for which there are long-term observations at this site.

Specific comments

Abstract Line 15: Suggest replacing "highly" with "strongly" here and throughout when referring to correlations

Line 15: "dry tracks" and "wet tracks" not yet defined. The correlates of nitrate reflected in the $R^2$ values are unclear.

Introduction The Introduction is long relative to the study's objectives and to other papers published in this journal. Hone in on documented factors that influence nutrient availability and potential links between dynamics of nutrients and carbon fluxes, and pare away ideas that do not directly inform the present analyses.

p. 2, line 30: delete one instance of Arctic

p. 2, lines 31-32: Increased specificity needed here with respect toe "projected increases." Does this refer to $CO_2$ flux?

p. 3, lines 4-5: This text is identical to the text of Commane et al. Commane et al. is not included in the Literature Cited section.

p. 3, lines 17-20: How are "high Arctic" and "wetlands" defined here? Many study sites cited as such are not classified by the original authors as wetlands or geographically within the high Arctc (e.g., alpine tundra in the Alaska range)

p. 4, line 5: define CBAWO

p. 4, line 28: revise for grammar

p. 6, line 14: Please describe the spacing of the points on the sampling grid.

p. 7, line 30: Please report limits of quantitation and how samples below these limits were handled.

p. 8, lines 16-17: Ecosystem respiration includes heterotrophic respiration, and therefore NEE-ER does not yield GPP. See Chapin et al. (2006) for consensus definitions of carbon cycling terms. Chapin, F. S., Woodwell, G. M., Randerson, J. T., Rastetter, E. B., Lovett, G. M., Baldocchi, D. D., et al. (2006). Reconciling carbon-cycle concepts, terminology, and methods. Ecosystems, 9(7), 1041–1050.

Methods p. 8, line 18: Regression and Pearson correlation analyses are duplicative and only one should be reported. If the coefficients are of interest and linear associations are expected, use regression.

Results p. 9, line 12: I am not certain of the interpretation of the epsilon terms reported here, but I believe they are associated with the deviation from the sphericity assumption of the rmANOVA. Typically those values are used to correct the final P-value. It is unclear whether corrected P-values are reported.

p. 9, line 15: I recommend leaving out the within/between subjects language in favor of more straightforward reporting of the ecological pattern captured by each term.

p. 12, line 19: These regression statistics would be easier to interpret if reported on the corresponding panels of figures 5 & 6. However, the regressions should be performed as multiple regressions to avoid inflating the chance of false positives. Further,

collinearity among predictors should be addressed.

p. 13, line 8: Referring to a table or figure, rather than a p-value to support the result would provide clarity.

p. 13, lines 30-32: Many correlational relationships are described here, and it would be fruitful to speculate about multiple potential causal associations. For example, seasonal patterns in these abiotic attributes might co-occur with the light regime and therefore NPP, resulting in less labile substrate to fuel ER, but with no direct effects of moisture, temperature, and active layer on ER.

p. 14, line 10: Has it been established that soils at the study site are anoxic?

p. 14, lines 17-20: There are some potentially interesting ideas about the drivers of N dynamics listed here, but the effectiveness of this discussion would be improved if the logic linking each of the factors was fully spelled out.

p. 14, lines 26-30: Here is another example of interpreting correlations as causal relationships. It is plausible that another factor, likely seasonality, drove both NPP and nitrate availability, rather than nitrate influencing GPP directly. Further, increased availability of nitrate in soils could occur due to lack of plant uptake of N.

p. 14-15, lines 33-2: The parallelism between the present and Toolik Lake result is not quite clear. How are patterns in light reflected in the present dataset?

p. 15, lines 19-20: Several papers from the Cape Bounty study have addressed nitrate dynamics from a catchment perspective. It seems relevant to place the present results into the existing context for the site.

p. 16, lines 10-28: Discussion of steps in the N cycle and nutrients (P) not addressed by the present dataset (available nitrate and ammonium) is beyond the scope of this study and detracts from its take-home messages.

Fig. 4: It would be helpful if the symbol colors or sizes were proportional to the resin

N content. Shapes could be used to represent wet/dry tundra. I don't think spatial interpolation is appropriate here because the area between the two sets of points is unsampled, and therefore error in the estimates varies greatly across the study area.

Tables 6-7: Interpretation of the B1, B2 identifiers is unclear.

Fig. 5: These plots require labels with units on both axes.

[Figure]

---

## Author Comment (AC1) · 25 Jan 2018

RC: General comments: Overall, this paper provides a clear method section and presents an extensive analysis of a dataset of available inorganic nitrogen in a high arctic wet sedge meadow. The spatial extent of data collection is welcome and given the importance of arctic wetlands, this study could provide a valuable addition from a different site. There is definitely value in improving understanding the controls on C and N cycling in arctic wetlands and there are potentially interesting data here. The manuscript, however, would benefit from an appropriate title, clearer aims and a greater effort to highlight what novel contribution the study makes. There is a substantial quantity of introductory text and results, and given that a fairly high proportion of the material confirms existing knowledge (such as the importance of hydrology in determining abiotic processes in wetlands, the importance of fine scale variability in Arctic ecosystems, the multiple environmental drivers of soil nutrient availability), a more concise summary of previous work would leave more room to explore novel findings. In view of the fact that the main issues are mostly to with approach to structure and content, I have provided comments in combination with the minor technical notes.

AC: We appreciate the insightful comments and questions posed by the reviewer here. Please see the attached supplementary for a revised abstract and more concise introduction. The title has also been changed to better reflect the paper's contents.

Specific and technical comments:

RC: Title and abstract - the abstract has a strong focus on N availability and fits well with the title, but a substantial proportion of the results and discussion relate to C exchange. Title and abstract should be modified to better reflect the content of the rest of the paper.

AC: This has been addressed in the attached supplementary. The title now reads "Available nitrogen and environmental controls on carbon exchange in a High Arctic wetland".

RC: The introduction would benefit greatly from a reduction in length, through taking a more focused approach to presenting the background information and rationale specific to this particular study. The current information is a sort of mini-review demonstrating the breadth of understanding of various elements of arctic ecosystem function but the text moves rapidly between wider issues such as global climate change (p3 lines 3-7), and the specifics of the study several times and this makes it challenging to disentangle exactly what the current understanding is and what gaps this study addresses. Specific sections - p2 line 16 to p4 line 6 includes multiple statements where the relevance isn't clear (for example, the lack of long-term studies in the Arctic, when this a short term

study), and p3 lines 20-30 could be summarized in much less text. The sentence at lines 4-6 is one of the few mentions of the key information underpinning this study and yet it is not stated what the past research is or what it showed.

AC: This has been addressed in the attached supplementary; the introduction has been shortened by a page from the original submission.

RC: P2, line 15 – 'this study looked at spatial patterns to see how these patterns shift through the growing season' – I presume what is meant is that spatial variability and temporal variability were investigated (doesn't make sense).

AC: Sentence removed; this has been addressed in the attached supplementary.

RC: P2, line 15 and elsewhere – the investigation of growing season variability seems overstated – is two sampling periods sufficient to investigate temporal patterns of nutrient availability, even in such a short season?

AC: All mentions of "growing season variability" will be replaced with "intra-seasonal variability", as we do understand that the use of the former is overstated in given the timeframe of this study, as it didn't capture the entirety of the growing season. Mentions of the study's timeframe will also be defined using the exact dates of study.

RC: P2, line 28 "global C changes" is vague – clarify what processes are being referred to here.

AC: Sentence should read ". . .and could contribute significantly to the global C balance."

RC: P4, line 5 – seems to be the first mention of the CBAWO wetlands, if so, define abbreviation.

AC: CBAWO should be defined as the "Cape Bounty Arctic Watershed Observatory (CBAWO)".

RC: P4, line 7 – not sure 'nutrient regime in plant growth' is the right phraseology for

what is meant here.

AC: The phrase has been removed.

RC: P4, line 25 – this sentence is unclear – what is meant by 'help in future predictions of decreasing carbon storage?'

AC: Sentence has been revised to read "Understanding the role N plays in C exchange in Arctic ecosystems will help in predicting the response of the Arctic C cycle to changes in temperature (Chapin et al., 2002; Mack et al., 2004)".

RC: P5, line 11 – what is a 'spatial lens'? Please clarify what exactly it is that hasn't been previously investigated.

AC: Sentence should read "However, the spatial relationships between available N and C exchange have not been assessed."

Methods

RC: Throughout methods paper refers to early and late season. This implies that the experiment took place either side of the 'mid' season, when actually it's a main growing season experiment that doesn't include 'early' and 'late' growing season (as acknowledged by the authors in the second paragraph of section 4.2). Suggest either using real time descriptions or clarifying in the methods section (2.2) where the sampled timed periods fit in the overall growing season.

AC: Addendums will be made to indicate the specific timing of the sampling periods in the overall growing season.

RC: P6, lines 4 and 5 – it is not clear how the vegetation communities differ between wet and dry tracks (i.e. do sedges and grasses dominate, with lesser elements of Salix arctica and herbaceous flowering plants in each of the different types, or does Salix arctica dominate in wet tracks?) With the genera (Carex and Eriophorum), can the authors say whether they're referring to two or more species (spp.) or one unspecified

species (sp.)? Also, is there a word (herbaceous?) missing between flowering and 'Eriophorum' (and if so, is this appropriate description as Eriophorum belongs to the Cyperaceae). I think Gramineae is now Poaceae. I appreciate that biotic controls are outside of the scope of this experiment, but there is a surprising lack of mention of the possible role of vegetation in any of the relationships described later, given that different species / plant functional types preferentially use different N sources (e.g. NH4, NO3, organic N).

AC: We agree that this needs to be clarified; the initial detail was not as clear as the biotic controls are outside of the experimental scope. Wet tracks were generally dominated by the members of the Cyperaceae family (i.e. Carex and Eriophorum spp.) and mosses (Sphagnum spp.). Dry tracks often had fewer graminoids (Poaceae, as indicated by the reviewer, not Gramineae), with Sphagnum spp. and various lichen genus underneath. Salix arctica was present in both wet and dry tracks, but less so in the dry tracks.

RC: P6, line 13 – alternate? (not alternative)

AC: The sentence should read "A total of 64 sites were established on alternating wet and dry strips sampled using ion exchange resin strips. . ."

RC: P6, line 15 – was? (not is)

AC: The sentence should read "The aim with this sampling design was to. . ."

RC: Results – is there a reason why this is structured in a different order from the methods? Consistency would make for an easier read.

AC: The order of sections will be reorganized in revision for better flow.

RC: P9, lines 3-5 – the comparison of data from this study with 2014 and 2015 comes out of nowhere – what is the importance of the June 2015 temperature to this study?

AC: Mentions of the 2014 and 2015 data to be removed, as they are not critical in this

study.

RC: P9 line 19, and P10, lines 5-8 and line 15 – where are the different tracks and what general concept does comparisons of difference between them relate to? This issue is common throughout much of the results (the spatial findings in particular) - references to what is happening on specific parts of the site by name rather than in context don't indicate to the reader how the findings of this study can be applied beyond this specific site.

AC: Please see supplementary Figure 1 for a graphic showing the spacing of the points on the sampling grid and location of wet and dry tracks in relation to each other; this will be included in the revised manuscript.

RC: P10, line 13 – incoming GPP?

AC: Sentence should read "As with ER, GPP was greater in the wet tracks than the dry tracks."

RC: P11, line 11 and P12, line 2 – opening sentences should be in the methods, not results

AC: Both sentences moved to Methods – Data Analysis section.

RC: P11, line 25 – this is a common finding with ion exchange resins in arctic ecosystems – saturation seems improbable when deploying the same resins in more nutrient rich ecosystems records often much higher totals, so could it be that in a longer burial some kind of equilibrium with soil levels is reached?

AC: The idea behind this sampling design of incorporating two different resin deployments was to compare the seasonality of the resins to test the technology in an Arctic setting. While that was not robustly designed in this paper, future study could look at nutrient additions to see if it is indeed an equilibrium with the soil levels that is reached.

RC: P11, line 30 – further investigation of what?

AC: Further investigation of ion exchange resin saturations levels or ideal burial lengths through experimental manipulations.

Discussion and conclusions

RC: These should be checked to see whether the data provide evidence to support all the assertions made, especially where they relate to processes that were assumed rather than measured - such as mineralization, nitrification and transport.

AC: We have reviewed the section and notes areas requiring clarification; assertions that are not directly backed by the data were taken out (i.e. references to mineralization reworded to reference NH4 availability, rather than direct mineralization, as that was not measured).

RC: Although there is discussion is context of other studies, the discussion is lacking in implications and does not clearly demonstrate what it is that this study shows that does not simply agree with previous findings (the majority). Where differences are highlighted (e.g. the presence of nitrate) no further thoughts are provided as to why this might be or what this will mean.

AC: The presence of nitrate alone in this system is notable, particularly for an aerobic process to occur in a waterlogged environment like an Arctic wetland. Different plants species utilize nitrate more efficiently than others (Smirnoff and Stewart, 1985; Nadelhoffer et al., 1991), hence the implications of this could influence future aboveground biomass composition and promote inter-species competition for nitrate-N.

RC: Substantial parts of section 4.2 are suggestions for other studies – although this is interesting, it doesn't relate to the sub-heading and it could be summarized in a couple of sentences, rather than providing a heavily referenced rationale.

AC: The original submission of Section 4.2 lacked much of the main findings from this study pertaining to comparison the strength of relationships between environmental variables in predicting carbon flux vs. inorganic N and environmental variables in

predicting carbon flux. These findings will be included in subsequent revisions of the manuscript. To summarize, as seen in the attached Figure 2, the strength and spread of the relationship to carbon dioxide exchange is tightened when plant-available nitrogen forms are factored in. To date, many studies of Arctic wetlands have not factored in the importance of soil-available nutrients in explaining seasonal variability of carbon flux.

RC: In the conclusions, some of them seem not arise from the results presented (for example, was there a test of the relationship between distance from snowpack on N availability, and is there any evidence that mineralization promotes photosynthetic activity?) and many of them are readily referenced to other older studies that it is not clear what has been found that is not already well known.

AC: References to the snowpack and spatial aspects relating to that will be taken out, as they are not critical to the study.

RC: Figs 5 and 6 – add y-axis labels.

AC: Figures edited to add y-axis labels

RC: Table 7 – title doesn't match table contents

AC: Titles for Table 7 and 9 should read "Bivariate regression R2 coefficients for carbon flux measurements (dependent variable) against nitrogen (independent variable)..."

Please also note the supplement to this comment:
https://www.biogeosciences-discuss.net/bg-2017-440/bg-2017-440-AC1-supplement.pdf

———————————————

[Figure]

**Fig. 1.** Natural cover image of the study area with wet (red) and dry (green) plots overlaid

[Figure]

**Fig. 2.** Multiple regression results using environmental variables and nitrogen in predicting CO2 exchange

**Supplement:**

**Available nitrogen and environmental controls on carbon exchange in a High Arctic wetland**

Jacqueline K.Y. Hung[1], David M. Atkinson[2], Neal A. Scott[1]

[revised manuscript text omitted]

Arctic wetlands also contribute significantly to the landscape-scale C balance in the High Arctic. These wetlands contain a disproportionate amount of soil C (Post *et al.*, 1982; Grogan and Chapin, 1999) and could contribute significantly to the global C balance if this carbon is released (Schuur *et al.*, 2008). It is expected that increased C losses from the terrestrial environment will result from permafrost degradation due to the changing global climate (Tarnocai *et al.*, 2009). Other research, however, has shown that High Arctic wetlands have the potential for increased productivity and C storage associated with changes in temperature and precipitation (Nobrega and Grogan, 2008; Hill and Henry, 2011). These potential increases in productivity can help offset possible soil C losses associated with permafrost thawing. Short-term studies of Arctic tundra environments suggest that Arctic permafrost regions currently act as sinks of atmospheric and terrestrial C (Nobrega and Grogan, 2008; Lafleur *et al.*, 2012; McGuire *et al.*, 2012); however, the comprehensive C study by McGuire *et al.* (2012) also determined that the tundra has been C neutral in recent decades. Generally, long-term C studies are still lacking across the High Arctic tundra (Euskirchen *et al.*, 2016). A long-term study by Euskirchen *et al.* (2016) found that increases in air and soil temperatures at multiple depths may trigger a new trajectory of $CO_2$ release. In addition, global climate models by Avis *et al.* (2011) have projected that permafrost degradation can lead to a decrease in the areal extent of wetlands, decreasing their contribution to the

landscape-scale C balance in the High Arctic. More research is needed to understand the relationship between climatic warming and Arctic wetland structure and function to determine their future contribution to climate warming.

The balance between ecosystem respiration (ER) and gross primary productivity (GPP) determines the net terrestrial C balance. Arctic wetlands have long been regarded as C sinks due to the dominance of GPP over ER (Mikan, *et al.*, 2002; Nobrega and Grogan, 2008; Lafleur *et al.*, 2012; McGuire *et al.*, 2012). However, warming temperatures are expected to increase ER in the Arctic, which could lead to a shift in the C balance depending on the response of GPP; this could lead to decreased net C storage and make permafrost regions net C sources (Welker *et al.*, 2004; Commane *et al.*, 2017). Belowground $CO_2$ release is season-dependent and strongly influenced by climate (Grogan and Chapin, 1999; Sullivan *et al.*, 2008), and High Arctic wetlands may respond more rapidly due to the limited temperature ranges of high latitudes. Studies at Toolik Lake, Alaska found that simulated warming made wet sedge tundra plots a weak sink for $CO_2$ at the peak of the growing season, but only for a short period of time (Johnson *et al.*, 2000). Similarly, a study that included a Canadian High Arctic wetland at Alexandra Fiord on Ellesmere Island showed that a wet sedge meadow was a net $CO_2$ source until switching to a sink closer to the end of the growing season (Welker *et al.*, 2004). Most of the previously mentioned studies did not consider how changes in nutrient availability might influence ecosystem C balance.

Changes in climate could also influence N cycling processes, with consequences for the net C balance (Chapin *et al.*, 2002; Mack *et al.*, 2004). It is important, therefore, to consider how climate change will alter microbial controls on N cycling. Mikan *et al.* (2002) found that warming in laboratory incubation studies stimulated microbial activity and increased nutrient availability in thawed soils. Microbial activity is known to remain active throughout the winter season, and can contribute significantly to nutrient budgets during spring thaw (Hobbie and Chapin, 1996; Schmidt and Lipson, 2004; Edwards *et al.*, 2006). Because the insulating snow layer prevents Arctic mid-winter soils from falling below -10°C (Clein and Schimel, 1995), the occurrence of freeze-thaw events allows microorganisms to remain active as long as pockets of liquid water are still present as a result of the snow insulation (Edwards *et al.*, 2006). The activity of these microorganisms will mobilize N stores that can help mitigate the current nitrogen limitation in High Arctic ecosystems (Mikan *et al.*, 2002), promoting plant production and soil respiration in the following growing season. In High Arctic wetlands, where microbial metabolism is primarily anaerobic due to the anoxic conditions, changes to drainage, precipitation, or evapotranspiration patterns will be the primary driver of microbial activity changes in the future (Mikan *et al.*, 2002; Zamin *et al.*, 2014).

Future warming could alter interactions between C and N cycles in the High Arctic. Warming will lead to a deeper active layer, lower water table, and promotion of subsurface flow pathways and water availability that enhances nutrient release from decomposing organic matter (Biederbeck and Campbell, 1973; Billings *et al.*, 1982; Nadelhoffer *et al.*, 1991; Johnson *et al.*, 2000; Natali *et al.*, 2011). These factors in turn affect other abiotic characteristics such as soil moisture (SM) and pH, which have a strong impact on both C and N cycling processes. High Arctic plant growth is typically limited by N availability (Nadelhoffer *et al.*, 1992; Shaver and Chapin, 1995; Shaver *et al.*, 2000). While Arctic plant growth is typically slow, relative to more productive, nutrient-rich temperate environments, tundra environments are more responsive to short-term (1-10 year) changes in nutrient availability (Shaver *et al.*, 2000). Due to the interconnected nature of the physical environment and microbial processes, it is expected that changes in nutrient

availability will influence C exchange with the atmosphere. In tundra microcosms (Billings *et al.*, 1984), increased soil N significantly increased $CO_2$ uptake. Decomposition of SOM as a result of warmer soils can release plant-available N forms, enhancing plant growth and ecosystem productivity (Hill and Henry, 2001; Weintraub and Schimel, 2005). Other research has demonstrated the importance of environmental variability on C exchange in High Arctic ecosystems (Blaser, 2016). This study explores the relationships between environmental variables and C and N cycling processes to determine the spatial and temporal extent of these relationships in the CBAWO wetlands.

In this study, we characterize the spatial and temporal patterns of available soil N in a wet sedge meadow in the Cape Bounty Arctic Watershed Observatory of the Canadian High Arctic, how they relate to environmental variability, and the consequences for C exchange processes. Many factors determine the levels of plant-available N that fuel N cycling and C flux (Nadelhoffer *et al.*, 1991; Clein and Schimel, 1995; Chapin *et al.*, 2002), and the need to understand the relationship between the environmental variables and these processes is more important than ever in a rapidly changing climate. Previous studies in this wet sedge meadow have looked at the seasonal variability of C exchange processes and environmental variables across wet and dry areas within wet-sedge meadows (Blaser, 2016). However, the relationships between environmental variables, available N, and C exchange have not been assessed.

---

## Author Comment (AC2) · 25 Jan 2018

RC: Hung et al. examined spatial heterogeneity in soil nutrient pools, effects of prospective abiotic drivers of nutrient availability, and relationships of nutrients and soil moisture with carbon balance of a High Arctic ecosystem. The study identifies landscape positions and times within the growing season that support strong links between nutrients and productivity. Empirical studies such as this have potential to reveal relationships between source-sink dynamics of carbon and spatial and temporal variation in soil moisture that have previously been unrecognized. This study quantifies correlational relationships among carbon fluxes, nutrient availability, and abiotic attributes of

soils and extends these correlations to assess mechanistic relationships. Greater caution and scrutiny must be applied to interpreting correlational relationships to consider alternative explanations that do not include direct causal links between the measured attributes. For example, coherent temporal patterns between nitrogen pools and productivity might result if both are responding to a shared driver, and might not reflect a direct effect of nitrogen on plant production. The manuscript's context is broad relative to the limited spatial and temporal extent of data collection. There is value in such focused studies, as they can reveal key patterns that might affect processes at larger scales (e.g., regional C balance), but the patterns revealed by the current analyses and their potential implications tend to get lost among discussions of tangential processes not directly addressed by the data in-hand (e.g., phosphorus, N transformations). Finally, there is a missed opportunity to compare patterns in soil nutrients with nitrogen dynamics at the watershed scale, for which there are long-term observations at this site.

AC: We appreciate the insightful comments and questions posed by the reviewer here. Please see the attached supplementary for a revised abstract and more concise introduction. The title has also been changed to better reflect the paper's contents.

Specific comments

Abstract

RC: Line 15: Suggest replacing "highly" with "strongly" here and throughout when referring to correlations

AC: The wording has been changed and this has been addressed in the attached supplementary.

RC: Line 15: "dry tracks" and "wet tracks" not yet defined. The correlates of nitrate reflected in the R2 values are unclear.

AC: This has been clarified in the attached supplementary.

Introduction

RC: The Introduction is long relative to the study's objectives and to other papers published in this journal. Hone in on documented factors that influence nutrient availability and potential links between dynamics of nutrients and carbon fluxes, and pare away ideas that do not directly inform the present analyses.

AC: The introduction has been shortened by a page from the original submission; the attached supplementary has the revised introduction.

RC: p. 2, line 30: delete one instance of Arctic

AC: Sentence revised to read "Preliminary research has predicted that Arctic wetlands have the potential to increase C outputs. . ."

RC: p. 2, lines 31-32: Increased specificity needed here with respect to "projected increases." Does this refer to CO2 flux?

AC: The sentence has been changed to clarify the meaning; the sentence and its preceding sentence now read "Preliminary research has predicted that Arctic wetlands have the potential for increased greening and productivity with increased temperatures and precipitation inputs (Nobrega and Grogan, 2008; Hill and Henry, 2011). These potential increases can help offset the projected increases of CO2 flux through C uptake during photosynthesis."

RC: p. 3, lines 4-5: This text is identical to the text of Commane et al. Commane et al. is not included in the Literature Cited section.

AC: This sentence was taken out as it is not critical to the manuscript. This has been addressed in the attached supplementary.

RC: p. 3, lines 17-20: How are "high Arctic" and "wetlands" defined here? Many study sites cited as such are not classified by the original authors as wetlands or geographically within the high Arctc (e.g., alpine tundra in the Alaska range)

AC: The sentence should be edited to just read "Arctic wetlands have long been re-garded as C sinks...". Most studies refer generally to Arctic wetlands; this study here looks at a wet sedge meadow, a type of wetland representative of Arctic wetlands. Ref-erences of studies in the Alaskan Range have been removed as they are not pertinent to this study. These changes are reflected in the attached supplementary.

RC: p. 4, line 5: define CBAWO

AC: CBAWO should be defined as the "Cape Bounty Arctic Watershed Observatory (CBAWO)".

RC: p. 4, line 28: revise for grammar

AC: Sentence should read "Microbial controls on nutrient cycling are important pro-cesses to consider in High Arctic environments."

RC: p. 6, line 14: Please describe the spacing of the points on the sampling grid.

AC: Please see supplementary Figure 1 showing the spacing of the points on the sampling grid and location of wet and dry tracks in relation to each other.

RC: p. 7, line 30: Please report limits of quantitation and how samples below these limits were handled.

AC: The limits of detection for the segmented flow analyzer used is sub-parts per billion; no samples went below that limit.

RC: p. 8, lines 16-17: Ecosystem respiration includes heterotrophic respiration, and therefore NEE-ER does not yield GPP. See Chapin et al. (2006) for consensus def-initions of carbon cycling terms. Chapin, F. S., Woodwell, G. M., Randerson, J. T., Rastetter, E. B., Lovett, G. M., Baldocchi, D. D., et al. (2006). Reconciling carbon-cycle concepts, terminology, and methods. Ecosystems, 9(7), 1041–1050.

AC: All instances of net ecosystem exchange (NEE) changed to net ecosystem pro-ductivity (NEP) as per Chapin et al., 2006.

Methods

RC: p. 8, line 18: Regression and Pearson correlation analyses are duplicative and only one should be reported. If the coefficients are of interest and linear associations are expected, use regression.

AC: Results from the Pearson correlation analyses will be removed for revisions; only regression analysis will be reported.

Results

RC: p. 9, line 12: I am not certain of the interpretation of the epsilon terms reported here, but I believe they are associated with the deviation from the sphericity assumption of the rmANOVA. Typically those values are used to correct the final P-value. It is unclear whether corrected P-values are reported.

AC: The corrected P-values were not reported as they were significant at all levels, so the standard P-value was shown.

RC: p. 9, line 15: I recommend leaving out the within/between subjects language in favor of more straightforward reporting of the ecological pattern captured by each term.

AC: This will be addressed in the revision; within/between subjects language will be substituted with between moisture tracks/across the season.

RC: p. 12, line 19: These regression statistics would be easier to interpret if reported on the corresponding panels of figures 5 & 6. However, the regressions should be performed as multiple regressions to avoid inflating the chance of false positives. Further, collinearity among predictors should be addressed.

AC: The original submission of the last paragraph of Section 3.6 and Section 4.2 lacked much of the main findings from this study pertaining to comparison the strength of relationships between environmental variables in predicting carbon flux vs. inorganic N and environmental variables in predicting carbon flux. These findings will be included in
subsequent revisions of the manuscript. To summarize, as seen in the attached Figure 2, the strength and spread of the relationship to carbon dioxide exchange is tightened when plant-available nitrogen forms are factored in. Collinearity among the predictors was not an issue

RC: p. 13, line 8: Referring to a table or figure, rather than a p-value to support the result would provide clarity.

AC: This statement is in reference to Figure 2 of the original submission.

RC: p. 13, lines 30-32: Many correlational relationships are described here, and it would be fruitful to speculate about multiple potential causal associations. For example, seasonal patterns in these abiotic attributes might co-occur with the light regime and therefore NPP, resulting in less labile substrate to fuel ER, but with no direct effects of moisture, temperature, and active layer on ER.

AC: The manuscript revision will include discussion on the multiple regression models that were explored and the multiple potential causal associations that these results present.

RC: p. 14, line 10: Has it been established that soils at the study site are anoxic?

AC: The redox potential has not been assessed for this study site, but saturated soils of this nature are generally anoxic.

RC: p. 14, lines 17-20: There are some potentially interesting ideas about the drivers of N dynamics listed here, but the effectiveness of this discussion would be improved if the logic linking each of the factors was fully spelled out.

AC: The links between the factors will be clarified and expanded in the revised manuscript.

RC: p. 14, lines 26-30: Here is another example of interpreting correlations as causal relationships. It is plausible that another factor, likely seasonality, drove both NPP

and nitrate availability, rather than nitrate influencing GPP directly. Further, increased availability of nitrate in soils could occur due to lack of plant uptake of N.

AC: The revisions will be made using more careful wording of the implications of the results (i.e. "suggests" instead of "indicates") to avoid misinterpretation of causal relationships.

RC: p. 14-15, lines 33-2: The parallelism between the present and Toolik Lake result is not quite clear. How are patterns in light reflected in the present dataset?

AC: The patterns in light are not reflected in the present dataset, although PAR measurements were taken (but not reported). The sentence should be rephrased to focus on the relationship between increased soil temperatures and higher ammonium availability; mentions of light attenuation will be taken out in revisions as they do not strengthen the argument.

RC: p. 15, lines 19-20: Several papers from the Cape Bounty study have addressed nitrate dynamics from a catchment perspective. It seems relevant to place the present results into the existing context for the site.

AC: Addendums will be made to the revisions to include discussion on catchment-wide nitrate dynamics (i.e. Louiseize et al., 2014; Lafrenière et al., 2017).

RC: p. 16, lines 10-28: Discussion of steps in the N cycle and nutrients (P) not addressed by the present dataset (available nitrate and ammonium) is beyond the scope of this study and detracts from its take-home messages.

AC: This section will be taken out in the revisions as it does not pertain to the study.

RC: Fig. 4: It would be helpful if the symbol colors or sizes were proportional to the resin N content. Shapes could be used to represent wet/dry tundra. I don't think spatial interpolation is appropriate here because the area between the two sets of points is unsampled, and therefore error in the estimates varies greatly across the study area.

AC: This figure will be taken out in the manuscript revision as it does not contribute significantly to the main findings.

RC: Tables 6-7: Interpretation of the B1, B2 identifiers is unclear.

AC: A, B1, and B2 are in reference to the total, early, and late season resin deployments; this will be clarified in the revised manuscript.

RC: Fig. 5: These plots require labels with units on both axes

AC: This change has been made for the revision.

Please also note the supplement to this comment:
https://www.biogeosciences-discuss.net/bg-2017-440/bg-2017-440-AC2-supplement.pdf

[Figure]

**Fig. 1.** Natural cover image of the study area with wet (red) and dry (green) plots overlaid

[Figure]

**Fig. 2.** Multiple regression results using environmental variables and nitrogen in predicting CO2 exchange